# Improving Generative Behavior Cloning via Self-Guidance and Adaptive Chunking

**Junhyuk So**[*1], **Chiwoong Lee**[*2], **Shinyoung Lee**[2], **Jungseul Ok**[1,2], **Eunhyeok Park**[1,2]
[1]Department of Computer Science & Engineering
[2]Graduate School of Artificial Intelligence
POSTECH, South Korea
{junhyukso,chiwoonglee,shinyoung,jungseul,eh.park}@postech.ac.kr

## Abstract

Generative Behavior Cloning (GBC) is a simple yet effective framework for robot learning, particularly in multi-task settings. Recent GBC methods often employ diffusion policies with open-loop (OL) control, where actions are generated via a diffusion process and executed in multi-step chunks without replanning. While this approach has demonstrated strong success rates and generalization, its inherent stochasticity can result in erroneous action sampling, occasionally leading to unexpected task failures. Moreover, OL control suffers from delayed responses, which can degrade performance in noisy or dynamic environments. To address these limitations, we propose two novel techniques to enhance the consistency and reactivity of diffusion policies: (1) self-guidance, which improves action fidelity by leveraging past observations and implicitly promoting future-aware behavior; and (2) adaptive chunking, which selectively updates action sequences when the benefits of reactivity outweigh the need for temporal consistency. Extensive experiments show that our approach substantially improves GBC performance across a wide range of simulated and real-world robotic manipulation tasks. Our code is available at `https://github.com/junhyukso/SGAC`.

## 1 Introduction

With the rapid advancement of generative models [1, 2, 3, 4] across a wide range of domains [5, 6, 7, 8], their adoption in robot learning is also accelerating [9, 10, 11, 12]. One compelling direction is Generative Behavior Cloning (GBC), which reinterprets the classic problem of Behavioral Cloning (BC) [13] using the modern generative models. In traditional BC, expert demonstrations, pairs of observed states and corresponding actions, are collected to train a model that maps observations to actions. GBC extends this idea by leveraging the strong generalization capabilities of state-of-the-art generative models to learn this mapping more effectively. Recent studies [14, 9, 10] show that GBC can handle complex sequential decision-making tasks across diverse environments using only supervised signals, greatly simplifies sample collection and training process without the need of intricate reinforcement learning.

Among recent trends, one particularly notable line of research is the Diffusion Policy model [9]. By adapting the score-based diffusion process originally developed for vision tasks, this approach enables sequential action generation through iterative refinement in a stochastic action space. This method has demonstrated significantly higher success rates compared to prior works [15, 16], representing a promising direction in BC. In particular, the integration of open-loop (OL) control, where a single observation is used to generate a sequence of future actions, combined with the powerful generalization capability of diffusion models, leads to improved temporal consistency, higher effective

---

[*]Equal contribution

39th Conference on Neural Information Processing Systems (NeurIPS 2025).

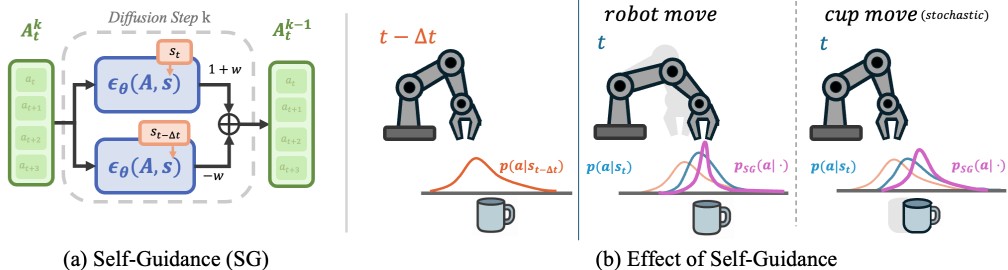

(a) Self-Guidance (SG)         (b) Effect of Self-Guidance

Figure 1: Illustration of our Self Guidance(SG). By using the past state distribution as negative guidance, SG effectively sharpens the distribution or proactively reacts to environmental perturbations.

control frequencies, and ultimately smoother, more stable motions with substantially better overall performance.

However, this approach also comes with inherent limitations. Owing to the stochastic nature of diffusion-based sampling, there remains a non-trivial risk of generating erroneous actions that can result in task failure. In OL control, even a single poor action can unfold over multiple consecutive time steps, leading to a significant drop in performance. Additionally, OL control lacks the ability to respond promptly to unexpected disturbances, making it particularly fragile in noisy or dynamic environments. Closed-loop (CL) control, where actions are generated at each time step based on real-time observations, offers a more reactive alternative. However, it introduces a different challenge: the difficulty of maintaining temporal consistency. Because diffusion models sample stochastically at every step, CL control often suffers from jittery or unstable behavior, which can severely degrade performance. These limitations raise two critical questions: *How can we increase the likelihood of sampling high-quality actions?* And *how can we achieve both reactivity and consistency while leveraging the strengths of diffusion policies?* Addressing these questions is essential for unlocking the full potential of diffusion-based decision-making systems in real-world applications.

In this study, we address these two fundamental challenges in diffusion-based control. First, we introduce a novel form of self-guidance that incorporates negative score estimates, derived from prior observations, into the diffusion denoising process. While diffusion guidance has been extensively studied in image generation to improve sample quality [17, 18, 19], its application to behavioral cloning remains largely unexplored, primarily due to the difficulty of defining reward signals for imitation learning [14]. By leveraging information already embedded in the model's past decision, our method guides the model toward more informed, high-fidelity action modes and enables forward-looking extrapolation, all without requiring additional fine-tuning, as shown in Fig. 1.

In addition to this, we introduce adaptive chunking, a control mechanism that updates action chunks only when the benefits of increased reactivity outweigh the need for temporal consistency. This strikes a dynamic balance between the responsiveness of closed-loop control and the stability of open-loop planning. By combining self-guidance with adaptive chunking, our method significantly improves the performance of standard Diffusion Policy and other baselines. Extensive evaluations across simulated and real-world robotic environments demonstrate that our approach outperforms Vanilla Diffusion Policy by 23.25% and the state-of-the-art BID by 12.27%, while reducing computational cost by a factor of 16.

## 2 Preliminary and Related Works

This paper explores methods to improve BC performance using diffusion policy [9]. We first introduce the fundamental principles behind diffusion models and GBC, and then provide a clear comparison between OL and CL control, emphasizing the strengths and limitations of each.

### 2.1 Diffusion Models

Diffusion Probabilistic Models (DPMs) [20] have emerged as a powerful generative framework, where data generation is modeled as a gradual denoising process starting from a pure Gaussian distribution. Instead of directly learning the data distribution, DPMs aim to learn the *transition*

from a noise prior $p_T(x)$ to the target data distribution $p_{\text{data}}(\mathbf{x})$. This generative process was first formalized by Denoising Diffusion Probabilistic Models (DDPM) [4]. In DDPM, the forward process $q$ is defined as a fixed Markov chain that incrementally corrupts the data by adding Gaussian noise with a variance schedule $\beta_t \in (0, 1)$ over time steps $t = 1, \dots, T$:

$$q(\mathbf{x}_t \mid \mathbf{x}_{t-1}) = \mathcal{N}\left(\mathbf{x}_t; \sqrt{1-\beta_t}\,\mathbf{x}_{t-1},\, \beta_t \mathbf{I}\right). \tag{1}$$

By leveraging the properties of Gaussian distributions, one can sample $\mathbf{x}_t$ at any timestep $t$ directly from the original data $\mathbf{x}_0$, without sampling all intermediate steps:

$$q(\mathbf{x}_t \mid \mathbf{x}_0) = \mathcal{N}\left(\mathbf{x}_t; \sqrt{\bar{\alpha}_t}\,\mathbf{x}_0,\, (1-\bar{\alpha}_t)\mathbf{I}\right), \tag{2}$$

where $\alpha_t = 1 - \beta_t$ and $\bar{\alpha}_t = \prod_{i=1}^t \alpha_i$ denote the accumulated noise schedule. The reverse process, which learns to recover clean data from noisy observations, is also modeled as a Gaussian distribution. It is parameterized as:

$$p_\theta(\mathbf{x}_{t-1} \mid \mathbf{x}_t) = \mathcal{N}(\mathbf{x}_{t-1}; \boldsymbol{\mu}_\theta(\mathbf{x}_t, t), \boldsymbol{\Sigma}_\theta(\mathbf{x}_t, t)). \tag{3}$$

Here, $\boldsymbol{\mu}_\theta(\mathbf{x}_t, t)$ and $\boldsymbol{\Sigma}_\theta(\mathbf{x}_t, t)$ are typically predicted by a neural network. In many implementations, the variance $\boldsymbol{\Sigma}_\theta$ is fixed to a predefined schedule (e.g., $\tilde{\beta}_t \mathbf{I}$), while the mean $\boldsymbol{\mu}_\theta$ is derived from a noise prediction network $\boldsymbol{\epsilon}_\theta(\mathbf{x}_t, t)$, trained to estimate the noise $\boldsymbol{\epsilon}$ from the noised input. This formulation allows the model to gradually denoise $\mathbf{x}_t$ over time, ultimately recovering a clean data from the noise distribution. DDPM [4] further demonstrates that predicting the injected noise $\boldsymbol{\epsilon}$ is equivalent to minimizing a reweighted variational lower bound of the data log-likelihood. This leads to a remarkably simple training objective:

$$\mathcal{L}_{\text{simple}}(\theta) = \mathbb{E}_{t \sim U[1,T], \mathbf{x}_0 \sim p_{\text{data}}, \boldsymbol{\epsilon} \sim \mathcal{N}(\mathbf{0}, \mathbf{I})} \left\| \boldsymbol{\epsilon} - \boldsymbol{\epsilon}_\theta(\sqrt{\bar{\alpha}_t}\mathbf{x}_0 + \sqrt{1-\bar{\alpha}_t}\boldsymbol{\epsilon}, t) \right\|^2. \tag{4}$$

This formulation trains the network $\boldsymbol{\epsilon}_\theta$ to recover the original noise from a noisy sample, effectively teaching it to reverse the diffusion process. Building upon this, Song et al. [21] showed that the reverse diffusion process can also be interpreted as solving a *Stochastic Differential Equation (SDE)* or an equivalent *Probability Flow Ordinary Differential Equation (PF-ODE)*:

$$d\mathbf{x} = \left[ \mathbf{f}(\mathbf{x}, t) - \frac{1}{2}g(t)^2 \nabla_\mathbf{x} \log p_t(\mathbf{x}) \right] dt. \tag{5}$$

Here, the drift term involves the *score function* $\nabla_\mathbf{x} \log p_t(\mathbf{x})$, which is approximated by a neural network $s_\theta(\mathbf{x}, t)$. When the generative task is *conditional*, for example, guided by class labels, text prompts, or environment states, the score network is trained to predict the *conditional score*, $s_\theta(\mathbf{x}, t \mid c) \approx \nabla_\mathbf{x} \log p_t(\mathbf{x} \mid c)$. This allows the model to generate samples from a *conditional distribution* $p(\mathbf{x} \mid c)$, enabling controllable generation tailored to various downstream tasks.

## 2.2 Diffusion Policy for Generative Behavior Cloning

With this understanding of diffusion-based generative modeling, we now explore how these principles can be applied to the domain of control through GBC. Let us consider a demonstration dataset $D = \{\tau_j\}_{j=1}^N$, where each trajectory $\tau_j = \{(s_t^{(j)}, a_t^{(j)})\}_{t=0}^{T_j-1}$ consists of a sequence of state-action pairs collected from human experts. In this work, we train diffusion policy model [9], aiming to learn an *implicit policy distribution* $p_\theta(a_t \mid s_t)$ instead of a deterministic mapping from states to actions. This distributional approach enables the model to capture the diversity in plausible actions of expert behaviors. The training is performed by maximizing the log-likelihood of expert actions under the learned policy, using the following BC loss:

$$\mathcal{L}_{BC}(\theta) = \mathbb{E}_{(s_t, a_t) \sim D}[\log p_\theta(a_t \mid s_t)]. \tag{6}$$

Specifically, at time step $t$, we denote the action chunk as $A_t = a_{t:t+H}$, where $d_a$ is the dimensionality of each action. The diffusion policy learns to model the distribution over such action chunks using the following training objective, which mirrors the denoising score matching loss of Eq.(4):

$$\mathcal{L}_{DP}(\theta) = \mathbb{E}_{(A_t, s_t) \sim D, \boldsymbol{\epsilon} \sim \mathcal{N}(0, I), k \sim U[1, K]} \left\| \boldsymbol{\epsilon} - \boldsymbol{\epsilon}_\theta(A_t^k, k, s_t) \right\|^2. \tag{7}$$

Here, $A_t^k = \sqrt{\bar{\alpha}_k} A_t + \sqrt{1 - \bar{\alpha}_k} \cdot \epsilon$ represents a noised version of the action chunk $A_t$ at diffusion step $k$, where the noise schedule follows standard DDPM notation: $\alpha_k = 1 - \beta_k$, and $\bar{\alpha}_k = \prod_{i=1}^{k} \alpha_i$. During inference, the model samples a full action chunk $a_{t:t+H} \sim p(a_{t:t+H}|s_t)$ conditioned on the current state. The first $h$ actions of this chunk are then executed without replanning. In this setting, $H$ is referred to as the *prediction horizon*, while $h$ is the *action horizon*. By learning to predict joint distribution over long-horizon action sequences, the diffusion policy inherently acquires *implicit long-term capabilities.*

## 2.3 Trade-off between Open Loop and Closed Loop Controls

We define CL control as the case where *action horizon* $h = 1$, and OL control as the case where $h = H/2$, typically $H = 16, h = 8$, following the diffusion policy convention [9]. OL control is inherently vulnerable to unexpected disturbances that may occur within its $h$-step execution window as the entire actions $a_{t:t+H}$ is generated based solely on the past state $s_t$. (E.g; It's corresponding to 0.25s in 30Hz control frequency) In contrast, CL control replans at every step ($h = 1$), allowing it to react immediately to sudden changes in the environment. However, this frequent regeneration often compromise long-term planning and disrupts consistency between consecutive actions. This limitations reflect an inherent trade-off between consistency and reactivity, which must be carefully balanced in control design.

Due to its importance, several studies have attempted to address this inherent trade-off. ACT Policy [22], for instance, proposes the use of Exponential Moving Average(EMA), which ensemble current and past predictions to enhance temporal consistency. Most recently, BID [23] proposes a test-time search strategy that samples multiple candidate actions and selects the optimal one using two criteria: (i) *backward coherence*, which prefers actions that are most consistent with the previously executed ones, and (ii) *forward contrast*, which favors candidates that differ significantly from those generated proposed by a separate 'negative' (i.e., undesirable) model. While BID yields respectable performance gains, it comes at the cost of significant computation and inference latency, due to the need to evaluate numerous candidates and maintain an auxiliary model during inference.

## 2.4 Limitations of Prior Score Guidance in Diffusion Control

While diffusion policies sample actions $a_t \sim p_\theta(a_t \mid s_t)$ based on the current state $s_t$, the *inherent stochasticity* of generative models introduces the risk of producing *low-fidelity samples*, that is, actions with low compatibility or likelihood under the given state. Fig. 2 (a) illustrates the distribution of action chunks generated by a *Vanilla Diffusion Policy* [9]. As shown in the Fig. 2, a non-negligible subset of samples exhibits ambiguous or intermediate behaviors. These low-fidelity actions lead to degraded task performance, as shown in Fig. 2 (c). This issue becomes even more critical in stochastic environments, where the agent must rapidly adapt to newly observed states $s_t$.

A useful analogy comes from text-to-image generation, where outputs may deviate from the intended prompt. In such settings, users can simply discard unsatisfactory images and regenerate new ones. However, in sequential control tasks, this kind of post-hoc selection is often infeasible. A single erroneous action during rollout can lead to *task failure*, making fidelity essential for reliable control.

How, then, can we sharpen the distribution to filter out low-probability samples and enable rapid adaptation to changing states? A widely adopted approach in the image generation domain is Classifier-Free Guidance (CFG) [18], which modifies the denoising score during the diffusion process to steer the model toward more desirable outputs. Specifically, CFG applies the following guidance:

$$\texttt{CFG} \; : \; \hat{\epsilon}_{new} \leftarrow (1 + w) \cdot \epsilon_\theta(x, s_t) - w \cdot \epsilon_\theta(x, \emptyset). \tag{8}$$

Here, $w \in [0, +\infty]$ is referred to as the *guidance scale* and $\emptyset$ denotes `null` (unconditional). Recall that the noise prediction $\epsilon_\theta$ in diffusion models is proportional to the score of the data distribution, i.e., $\epsilon_\theta(\mathbf{x}_t, t, c) \propto \nabla_{\mathbf{x}_t} \log p_t(\mathbf{x}_t|c)$. Under this formulation, the modified score leads to a sampling distribution of the form $p_\theta(a|s_t) \cdot (p(a|s_t)/p(a))^w \propto p_\theta(a|s_t) \cdot (p(s_t|a))^w$, where the original distribution is effectively reweighted by a reward signal—namely, the classifier probability $p(s_t \mid a)$. Although this guidance mechanism has proven effective in the image generation domain [18], we observe that it does not translate well to sequential decision-making tasks, as demonstrated in Fig. 2(c), and similarly reported in prior work [14].

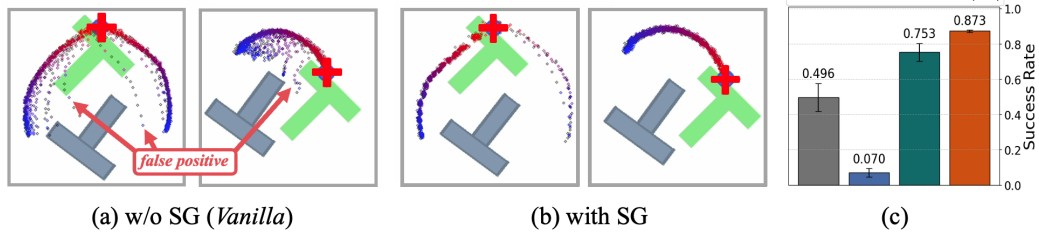

| | (a) w/o SG (*Vanilla*) | | (b) with SG | (c) |

Figure 2: (a) Visualization of the action distribution of Diffusion Policy (DP) [9]. (b) The sharpened distribution after applying our Self-Guidance(SG). (c) Their respective performances. Standard DP often generates low-fidelity actions, which can harm sequential control performance.

Another alternative is AutoGuidance (AG) [19], which replace the unconditional output used in CFG with a *conditioned output from an undertrained checkpoint*, denoted as $\epsilon_{\theta'}(x, s_t)$. This method builds on the insight that CFG's score modification can be interpreted as an *extrapolation* away from the output of a negative or 'bad' model, thereby enhancing the desired 'good' distribution [19]. The modified score of AG is computed as:

$$\texttt{AG} \; : \; \hat{\epsilon}_{\text{new}} \leftarrow (1 + w) \cdot \epsilon_\theta(x, s_t) - w \cdot \epsilon_{\theta'}(x, s_t). \tag{9}$$

As shown in Fig. 2 (c), AG significantly improves performance, highlighting the importance of filtering out false-positive actions. However, despite its effectiveness, AG has several limitations: (i) AG requires an additional checkpoint, doubling storage requirements; (ii) it relies on two separate model weights ($\theta$ and $\theta'$), which requires computing both noise predictions in multiple inferences; (iii) the selection of the 'bad' checkpoint $\theta'$ introduces an additional hyperparameter.

# 3 Methods

Motivated by the trade-offs and overhead observed in prior approaches, we present two novel methods that simultaneously improve reactivity and consistency in GBC, without requiring extra training or architectural changes. These techniques are designed to be lightweight and plug-and-play, making them easy to integrate into existing frameworks while delivering significant performance gains.

## 3.1 Self Guidance: Improving Fidelity and Reactivity of Diffusion Policy

Departing from prior methods that rely on auxiliary models or handcrafted guidance signals, we introduce a self-guided mechanism that is simpler, more efficient, and surprisingly more effective. Rather than introducing external guidance sources as in previous work, we propose a novel self-referential strategy that conditions on the model's own recent outputs—eliminating the need for extra models, tuning, or compute. Specifically, our Self Guidance (SG) is formulated as follows:

$$\texttt{SG} \; : \; \hat{\epsilon}_{new} \leftarrow (1 + w) \cdot \epsilon_\theta(x, s_t) - w \cdot \epsilon_\theta(x, s_{t-\Delta t}). \tag{10}$$

All that is required in SG is a single batched inference pass, using a concatenated conditioning input composed of the current and past states, $[s_t, s_{t-\Delta t}]$. This simple design makes SG highly efficient in both implementation and runtime, which is especially advantageous in resource-constrained scenarios.

For a more comprehensive understanding of SG, we provide a deeper analysis of its sampling behavior. Similar to CFG, SG modifies the sampling distribution as follows:

$$p_{\text{new}}(a) \propto p_\theta(a_t|s_t) \cdot \left( \frac{p_\theta(a_t|s_t)}{p_\theta(a_t|s_{t-\Delta t})} \right)^w. \tag{11}$$

This formulation encourages the model to assign higher probabilities to actions that deviate from those conditioned on the past state $s_{t-\Delta t}$, effectively guiding the model to adapt more rapidly to the newly observed state $s_t$.

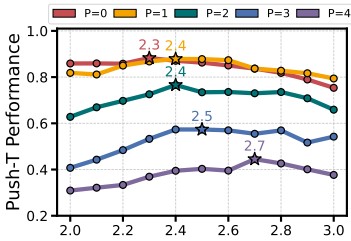

Figure 3: Effect of SG guidance scale($w$) on varying noise levels (P)

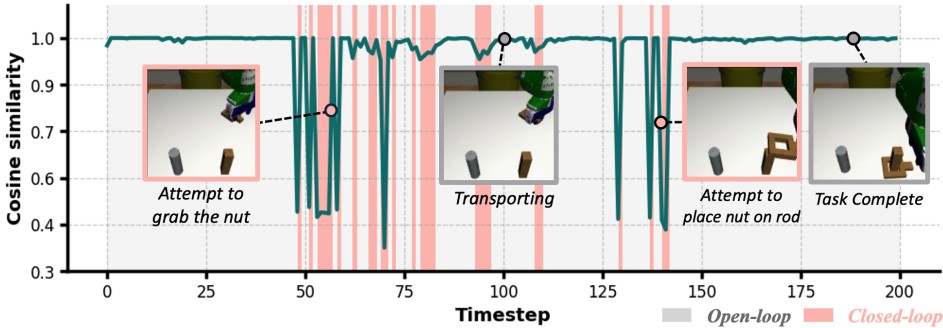

Figure 4: Similarity between actions from a previously planned chunk and newly replanned actions at each time step. The similarity tends to be high during simple movements (e.g., moving , transporting). Conversely, it tends to be low when high precision is required (e.g., attempting to grasp).

To give qualitative validation, we analyze how guidance strength affects overall performance under varying levels of stochasticity. In Fig. 3, the x-axis denotes the guidance weight $w$, while the y-axis shows the average final reward over 100 episodes. As the level of stochasticity increases, the optimal guidance weight rises accordingly—indicating that stronger guidance is beneficial under greater uncertainty. Notably, even in the absence of injected noise, SG significantly outperforms the vanilla setting ($w = 0$), demonstrating its effectiveness.

To deepen our understanding of the SG mechanism, we present an additional theoretical perspective based on temporal extrapolation. Under this view, Eq. 10 can be rewritten as:

$$\hat{\epsilon}_{new} \leftarrow (1 - w) \cdot \epsilon_\theta(x, s_t) + w \cdot (2 \cdot \epsilon_\theta(x, s_t) - \epsilon_\theta(x, s_{t-\Delta t})) \tag{12}$$
$$\simeq (1 - w) \cdot \epsilon_\theta(x, s_t) + w \cdot (\epsilon_\theta(x, s_{t+\Delta t})). \tag{13}$$

Assuming $\Delta t$ is small and $\epsilon_\theta(x, s)$ is locally smooth and differentiable with respect to the state $s$, the term $2 \cdot \epsilon_\theta(x, s_t) + \epsilon_\theta(x, s_{t-\Delta t})$ can be interpreted as a first-order approximation of $\epsilon_\theta(x, s_{t+\Delta t})$, with higher-order terms $\mathcal{O}((\Delta t)^2)$ being negligible. With this interpretation, the guidance mechanism effectively encourages the model to sample from a modified distribution: $p_{new}(a_t) \propto p_\theta(a_t|s_t)^{1-w} \cdot p_\theta(a_t|s_{t+\Delta t})^w$, which represents a weighted blend between the current state $s_t$ and an extrapolated future state $s_{t+\Delta t}$. This allows the model to generate actions that implicitly anticipate short-term future dynamics, thereby improving its ability to *adapt rapidly* and *respond proactively* to changes or disturbances in the environment.

### 3.2 Adaptive Chunking : Improving Consistency while Reactive

In addition to SG focusing on improving reactivity through more adaptive sampling, we now turn our attention to another key challenge in sequential control: maintaining temporal consistency without sacrificing responsiveness.

Due to its stochastic nature, diffusion policy tends to be less compatible with CL control, often exhibiting issues such as jittering or idling. On the other hand, the main limitation of OL control is its lack of reactivity, which leads to significant performance degradation in noisy environments.

Importantly, the effectiveness of each control mode depends heavily on the characteristics of the target operation. For tasks that require delicate and precise actions, such as grasping an object, the acceptable action space is narrow, and motor deviations must be minimal. In such cases, the instability typically associated with CL control is negligible, while its reactivity provides a clear advantage in responding to external disturbances. Conversely, for tasks involving large-scale movements, such as transporting or lifting an object, the action space is broader, and step-by-step replanning in CL control can introduce unnecessary acceleration changes, often leading to task failure. In these scenarios, OL control is more stable and preferable. Therefore, both CL and OL control offer distinct advantages depending on the context, highlighting the need for action-aware adaptive control strategies.

**Adaptive Chunking** Based on this observation, we propose an adaptive chunking method that selectively maintains open-loop execution when consistency is high, and reverts to closed-loop

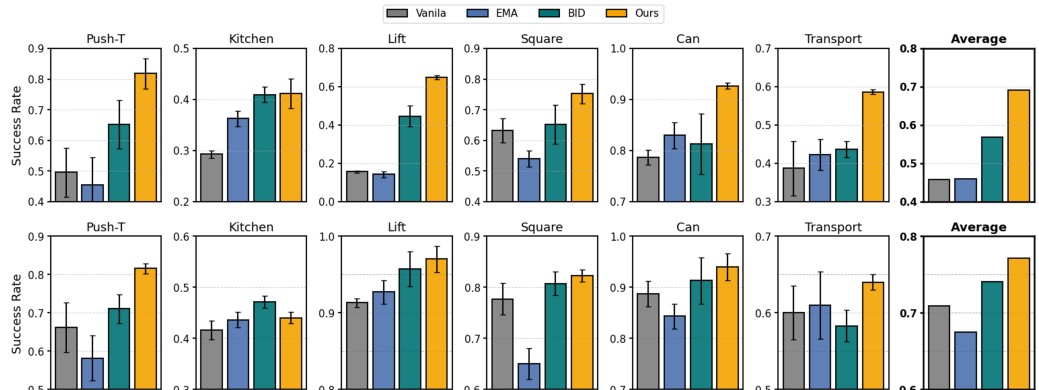

Figure 5: **Simulation Experiments :** *Stochastic***(top)** & *Static***(bottom)** : Performance comparison in the 6 simulated environment. Results are averaged over 100 episodes across three random seeds.

| Method | P=0 | P=1 | P=2 | P=3 |
|---|---|---|---|---|
| *Vanilla* [9] | 0.496 ± 0.080 | 0.322 ± 0.043 | 0.231 ± 0.032 | 0.204 ± 0.023 |
| EMA [22] | 0.456 ± 0.089 | 0.306 ± 0.012 | 0.229 ± 0.030 | 0.204 ± 0.018 |
| BID [23] | 0.652 ± 0.079 | 0.454 ± 0.029 | 0.236 ± 0.017 | 0.200 ± 0.020 |
| **Ours** | **0.819 ± 0.049** | **0.718 ± 0.026** | **0.413 ± 0.028** | **0.261 ± 0.017** |

Table 1: Comparison under different levels of stochasticity. The performance are evaluated on Push-T task and average over 100 episodes across 3 random seeds.

control when reactive updates are needed. Specifically, the model continues to use a previously planned action chunk as long as the similarity between the first action in the chunk and the newly generated action remains above a certain threshold.

Let $A_{\text{queue}}$ denote the action chunk queue, $\hat{a}_{t:t+H} \sim \pi(a \mid s_t)$ the newly predicted action chunk, and $\tau$ the similarity threshold. The update rule is defined as:

$$A_{queue} \leftarrow \begin{cases} A_{queue}.\texttt{enqueue}(\hat{a}_{t+H}) & \text{if } cos(A_{queue}[0], \hat{a}[0]) \geq \tau \\ \hat{a}_{t:t+H} & \text{else,} \end{cases} \tag{14}$$

where $cos(\cdot)$ denotes cosine similarity. At each timestep, the first action in the queue is dequeued and executed: $a_t = A_{queue}.\texttt{dequeue()}$.

This adaptive strategy enables the controller to operate in a *closed-loop fashion* during high-precision phases and switch to *open-loop execution* when exact actions are less critical. As a result, it effectively mitigates compounding errors while avoiding the typical problems of closed-loop control such as jittering and idling. Fig. 4 illustrates the similarity between actions from a previously planned chunk and newly replanned actions, along with the corresponding control mode selected by our adaptive chunking scheme. By dynamically selecting the appropriate control mode based on the execution phase, our method achieves significantly higher success rates across a variety of scenarios.

## 4 Experiments

To validate the effectiveness of our proposed method, we conduct experiments across various tasks and environments, ranging from simulation benchmarks to real-world applications. Moreover, we perform extensive ablation studies to investigate the impact and performance contributions of the different components integrated into our approach.

### 4.1 Simulation Experiments

We first evaluate the performance of our method on behavioral cloning tasks within six simulation environments. These include simple tasks like PushT [9], standard benchmarks from Robomimic [24], and the particularly challenging long-horizon Kitchen [25] environment. Success Rate is used for

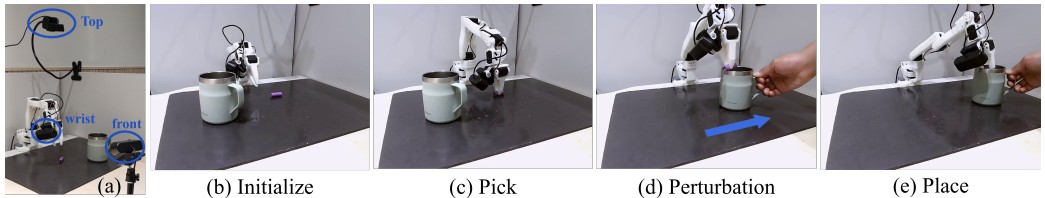

|  | (b) Initialize | (c) Pick | (d) Perturbation | (e) Place |

Figure 7: Real world experiment. (a) Experimental setup (b)-(e) Pick-and-place example.

main metric for most tasks, except for Push-T, which used target area coverage. For fair comparison, we endeavor to follow the evaluation setups of [23], with the primary modification being the use of a DDIM-30 [26] solver instead of the DDPM-100 [4] solver employed in the original work. Detailed setup configurations and results obtained using other solvers are included in the supplementary.

**Baselines** To demonstrate the effectiveness of our method, we conducted experiments comparing it not only against the *Vanilla* Diffusion Policy [9] but also against two other inference methods :

- **Exponential Moving Average (EMA)**: Introduced in [22], which also called temporal ensembling. During inference, actions are mixed with the previous action using a ratio $\lambda$: $\hat{A}_t = \lambda \cdot A_{t-1} + (1 - \lambda) \cdot A_t$ to enhance action smoothness. We set $\lambda = 0.5$.
- **Bidirectional Decoding (BID)** [23]: A state-of-the-art inference method for behavioral cloning that employs heavy test-time-search to select the optimal action sequence for a given state. We follow the default settings proposed in the original BID for fair comparison. Please refer to [23].

**Problem Setup** We consider two distinct problem setups as follows:

(i) **Stochastic**: Following [23], we introduce temporally correlated action noise during the manipulation task execution to simulate actuator noise or external disturbances. In this setting, *closed-loop* control is employed for fair comparison.

(ii) **Static**: We assume an ideal, clean environment without any external disturbances or noise. In this setting, *open-loop* control is utilized for all methods.

**Results** Fig. 5 illustrate the performance of our method compared to baselines in both *stochastic*(top) and *static*(bottom) cases. As shown, while EMA often improves performance, it results in performance degradation on some tasks. In contrast, both BID and our method consistently enhance performance across the evaluated simulation environments. However, BID not only worse than ours in performance but also it requires significant computational overhead - 16x more FLOPs and 2x slower latency. Our method, conversely, achieves superior performance than *Vanilla* DP by 23.25% and BID by 12.27%—without incurring additional computational cost.

## 4.2 Real World Experiments

Figure 6: Param. Sensitivity of EMA and AC.

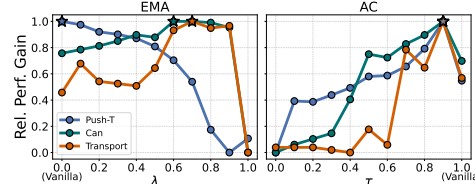

We further validate the practical applicability of our method through real-world experiments. Specifically, We utilized a Lerobot(Huggingface) [27] implementation of Diffusion Policy(DP), and deployed it on SO-100 low cost robot arm [28]. We employ 3-camera setup, top (bird's-eye), front, and wrist views visualized in Fig. 7(a). All experiments are conducted on one A6000 GPU server with DDIM-10 Solver with 30Hz standard visuomotor control frequencies.

**Problem Setup** We design simple pick-and-place task using pen holder and cup. The task involved picking up a pen-holder grip and placing it into a cup, as shown in Fig. 7. Similar to Sec. 4.1, we evaluated performance under two conditions: (i) *Stochastic* : The target cup is moved during task execution to introduce environmental disturbance; (ii) *Static* : The target cup remained stationary.

**Results** In Fig. 9, we report the Success Rate of *Vanilla* DP[9] and **Ours** accross 20 trials. As shown in Fig. 9, our method demonstrated stronger performance than the *vanilla*, especially in dynamic environments. This confirms the effectiveness and robustness of our approach beyond simulation and

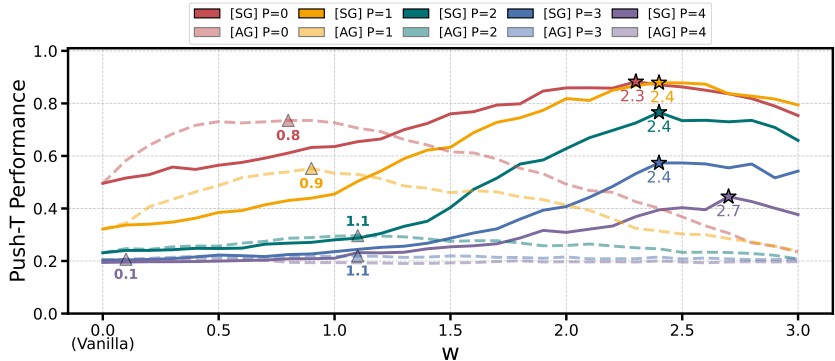

Figure 8: Effect of SG & AG guidance scale($w$) on varying noise levels (P)

its applicability to real-world scenarios with noisy hardware and potential disturbances. Moreover, while BID [23] shows halting behavior

### 4.3 Ablation Studies

**Different Levels of Stochasticity** Table 1 presents the performance results under varying environmental noise scale ($P$) on PushT task. The detailed experimental setup is in appendix. As shown, the performance of baselines degrades rapidly as noise scale increases. In contrast, ours maintains performance and outperforms other methods, showcasing effectiveness of Self Guidance's reactivity enhancement and robustness of Adaptive Chunking.

**Comparison with AutoGuidance [19].** To highlight the superior performance of our method, we conduct a detailed comparative study between our Self-Guidance (SG) and AutoGuidance (AG) [19]. In Fig. 15, we plot the performance of both methods across different guidance scales, ranging from $w = 0$ (no guidance) to $w = 3$, in various environment noise levels ($P$). As shown, while both methods improve performance with guidance, our SG consistently achieves a higher peak performance than AG across all evaluated noise levels. Moreover, AG's performance degrades rapidly as the noise scale increases, whereas our SG maintains its robustness even in noisy environments. Finally, AG introduces a significant computational burden, including storage costs for the weak model's weights and an increased effective latency due to inability due to an inability to perform. In contrast, our SG incurs no computational overhead while delivering superior performance.

**Sensitivity Analysis** While EMA [22] can achieve good performance with an optimal decay rate, it is often overly sensitive. In Fig. 6, we present a parameter sensitivity analysis comparing EMA's decay rate $\lambda$ with the threshold $\tau$ of our Adaptive Chunking (AC). As shown, EMA exhibits significantly different optimal decay rates across tasks. In contrast, our AC demonstrate consistent performance trends, highlighting their notable hyperparameter robustness and real-world applicability.

**Individual effect of SG and AC** In Fig. 10, we present an ablation study evaluating the impact of applying only Self Guidance (SG), only Adaptive Chunking (AC), and both components (Ours). As shown, the results indicate that while using either SG or AC alone improves performance over the baseline, the combination of both yields the best results.

## 5 Discussion : Extension to VLAs

While we mainly demonstrated our method with diffusion policy, our method can be extended to any behavior cloning framework that utilizes action chunking and probabilistic modeling of action space. To further validate the effectiveness and generality of our approach, we conducted experiments with two modern, state-of-the-art Vision-Language-Action models (VLAs) : $\pi_0$ and OpenVLA-OFT.

$\pi_0$**(pi-zero)** [12] is a recently proposed, state-of-the-art VLA pretrained on web-scale data, which demonstrates the potential of leveraging the embedded world knowledge of foundation models for general-purpose robotic planning and control. Specifically, it employs an early-fusion approach to process multimodal inputs and directly generates chunked actions through a denoising process. We integrated our SG method directly into this denoising stage also with AC.

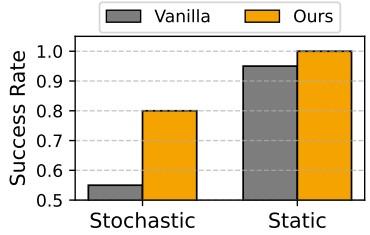

Figure 9: **Real World Experiments** We compare Success Rate(%) between Vanilla [9] and *Ours* under *stochastic* and *static* scenarios.

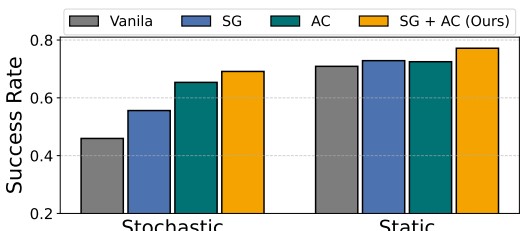

Figure 10: **Ablation study for our methods :** We depict individual performance of *ours* with average success rate across 6 simulation benchmarks.

**OpenVLA-OFT** [29] is another web-scale, fine-tuned Vision-Language-Action (VLA) model designed for robotic tasks, which also utilizes action chunking(OL) for control. However, since OpenVLA-OFT is not diffusion-based, our original SG method cannot be directly applied. To address this, we introduce an variant of our SG, inspired by recent LLM guidance techniques, activation steering [30].

Specifically, During the forward computation of $i$-th Transformer blocks $T^i$, we inject negative guidance using past activation $A_{t-1}^i$, as follows :

$$A_t^{i+1} \leftarrow T^i(A_t^i) + w \cdot (T^i(A_t^i) - T^i(A_{t-1}^i)). \quad (15)$$

This formulation in Eq. 15 is analogous to SG (Eq. 10) where we now applies guidance in feature-space instead of denoising output space in diffusion.

| Method | P=1 | P=5 |
|---|---|---|
| $\pi_0$ [12] | 82.0% | 12.2% |
| *closed-loop* | 73.4% | 14.4% |
| **Ours** | **83.8%** | **19.9%** |
| OpenVLA [29] | 82.0% | 12.2% |
| *closed-loop* | 73.4% | 14.4% |
| **Ours** | **83.8%** | **19.9%** |

Table 2: Performance of OpenVLA on LIBERO with different noise scale $P$

**Experimental Results** We use the LIBERO-Spatial benchmark [31] to evaluate performance. Similar to Sec. 4.2, we adopt a stochastic environment where target objects are in motion. Detailed experimental settings are provided in the Appendix. In Table 2, we compare the performance of original $\pi_0$ [12] and OpenVLA-OFT [29], which executed on *open-loop* control, and its *closed-loop* variant, finally with **ours**. As shown, the performance of vanilla VLAs with *open-loop* control decreases significantly in a stochastic environment (large $P$). While the closed-loop version shows some improvement in high-stochasticity regions, this improvement is marginal. In contrast, $\pi_0$ and OpenVLA-OFT combined with **our** method achieves the best performance across all tasks, highlighting its broad applicability and potential for future extensions to VLA-style models.

# 6 Conclusion and Limitations

In this work, we demonstrate that Generative Behavior Cloning, particularly Diffusion Policy, can suffer from low-fidelity issues and a reactivity-consistency trade-off. To address these, we propose two novel techniques: Self-Guidance, which injects past score predictions as negative guidance, thereby enhancing fidelity and reactivity; and Adaptive Chunking, which dynamically balances reactivity and consistency. Our experimental results show that our approach consistently improves robotic control quality across diverse scenarios, including both simulation and real-world applications.

**Limitations** One limitation of adaptive chunking is its computational cost, which is comparable to that of CL control due to step-wise similarity evaluations. Nevertheless, we view this as a valuable opportunity for future work, and believe that designing more computationally efficient similarity measures could further enhance the practicality of adaptive chunking.

# Acknowledgment

This work was supported by IITP and NRF grant funded by the Korea government(MSIT) (No. RS-2019-II191906, RS-2023-00213611, RS-2024-00457882).

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

# A   Experimental Details

**Hyperparameter Settings**   The hyperparameters used in our simulation experiments in main paper are summarized in Table. 3.

| Hyperparameter | Value |
|---|---|
| Observation history | 2 |
| Prediction horizon ($H$) | 16 |
| Diffusion Scheduler | DDIM-30 |
| $\tau$ of Adaptive Chunking | 0.97 (CL), 0.99 (OL) |
| $\Delta t$ of SG's past state $s_{t-\Delta t}$ | 1 |
| $w$ for SG [*Stochastic*] | 2.23 (Push-T) |
| | 0.86 (Square) |
| | 1.5 (Lift) |
| | 1.3 (Can) |
| | 1.17 (Transport) |
| | 0.62 (Kitchen) |
| $w$ for SG [*Static*] | 4.42 (Push-T) |
| | 1.25 (Square) |
| | 1.29 (Lift) |
| | 0.8 (Can) |
| | 1.02 (Transport) |
| | 0.64 (Kitchen) |

Table 3: Additional hyperparameters for simulation experiments.

**Implementation of Perturbation $P$**   In $P$ noisy environment setting, , we implement the disturbance by moving the T-block in a fixed direction at a velocity of $P$, which is the same implementation used in BID [23]. The goal of this stochastic scenario is to approximate environmental disturbances, such as slipperiness or wind.

# B   Experimental Results with DDPM-100 Solver

Fig. 3(main) reports the success rates for six tasks evaluated under two environments (*Stochastic* & *Static*) using the DDIM-30 solver [26]. To ensure a fair comparison with prior works [9, 23], we also visualize the results using the DDPM-100 solver [4], keeping all other hyper-parameters unchanged. As shown in Fig. 11, our method outperforms Vanilla Diffusion Policy [9] by **19.63%** and BID [23] by **7.58%** on average across the six tasks in the stochastic setting. In the static environment, our method still achieves higher performance, surpassing Vanilla Diffusion Policy by **1.90%** and BID by **1.23%**.

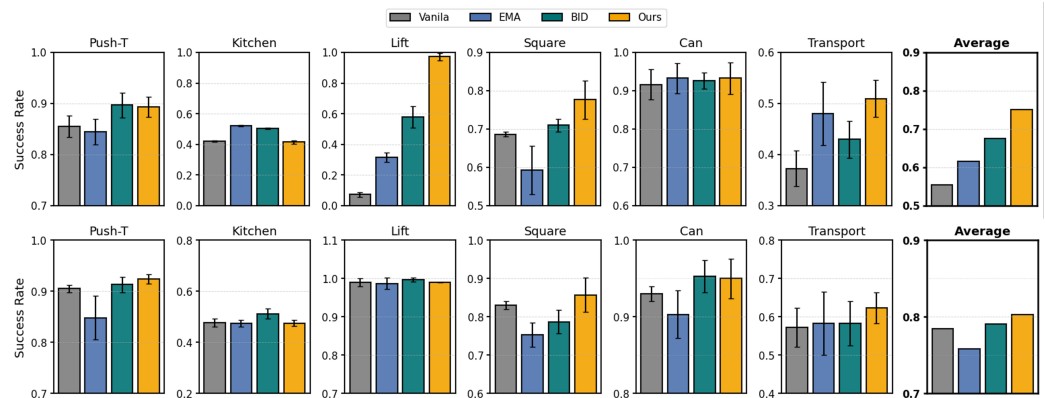

Figure 11: **Simulation Results with DDPM-100 solver :** *Stochastic***(top)** & *Static***(bottom)** : Performance comparison in the 6 simulated environment. Results are averaged over 100 episodes across three random seeds.

## C  Comparison with different EMA Rate $\lambda$

BID [23] reports that an Exponential Moving Average(EMA) [22] can perform well on several tasks, but also that the result is highly sensitive to the decay rate $\lambda$, with the optimal value differing by task. Motivated by this, we evaluated EMA over $\lambda \in \{0.0, 0.1, \ldots, 1.0\}$ for every task. Fig. 12 shows results for the *Stochastic* setting using representative values $\lambda \in \{0.1, 0.3, 0.5, 0.7, 0.9\}$, which include the empirically optimal value for each task. As shown Fig. 12, our method surpasses EMA on most benchmarks, highlighting both the challenge of choosing appropriate $\lambda$ for EMA and the robustness of ours.

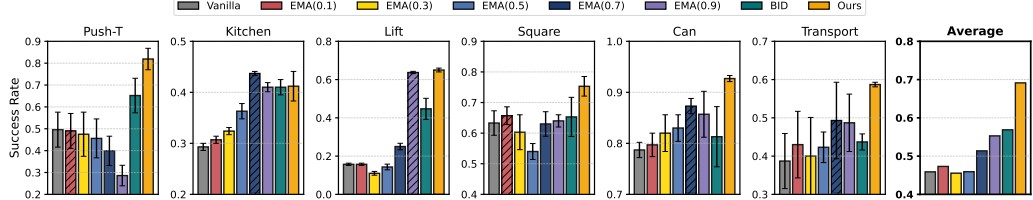

Figure 12: **Simulation Results on the Effect of $\lambda$ in EMA:** Performance comparison in the 6 simulated environment. Results are averaged over 100 episodes across three random seeds. The hatched bars indicate the optimal $\lambda$ value for EMA in each task.

## D  Comparison with different similarity Metric in Adaptive Chunking

To investigate effect of other similarity metrics in Adaptive Chunking(AC), we replaced $\cos(\cdot)$ with the $L_1$ and $L_2$ distances in AC. A threshold of $\tau = 0.1$ performed best for norm-based metrics, while $\tau = 0.97$ is used for the cosine-based method. As shown in Table 4, cosine similarity achieves the best performance across all tasks.

## E  Similarity visualization in other tasks

To verify the generality of our observation in Sec.3.2 of main, we also visualized the similarity of actions with *Can* task in Fig. 13.

Similar to [Fig.4 of main], the similarity decreases noticeably during precise actions in the *Can* task, such as grasping or placing the object into the target bin.

| | Vanilla | L1 | L2 | Cosine (Ours) |
|---|---|---|---|---|
| Push-T | $0.496 \pm 0.08$ | $0.621 \pm 0.012$ | $0.639 \pm 0.03$ | $\mathbf{0.72 \pm 0.037}$ |
| Kitchen | $0.298 \pm 0.007$ | $0.309 \pm 0.008$ | $0.323 \pm 0.014$ | $\mathbf{0.398 \pm 0.032}$ |
| Lift | $0.157 \pm 0.006$ | $0.227 \pm 0.015$ | $0.53 \pm 0.07$ | $\mathbf{0.587 \pm 0.006}$ |
| Square | $0.633 \pm 0.04$ | $0.583 \pm 0.015$ | $0.533 \pm 0.065$ | $\mathbf{0.753 \pm 0.021}$ |
| Can | $0.787 \pm 0.015$ | $0.687 \pm 0.068$ | $0.7 \pm 0.04$ | $\mathbf{0.91 \pm 0.00}$ |
| Transport | $0.387 \pm 0.07$ | $0.25 \pm 0.03$ | $0.25 \pm 0.053$ | $\mathbf{0.553 \pm 0.021}$ |

Table 4: Performance comparison of different vector metrics across tasks

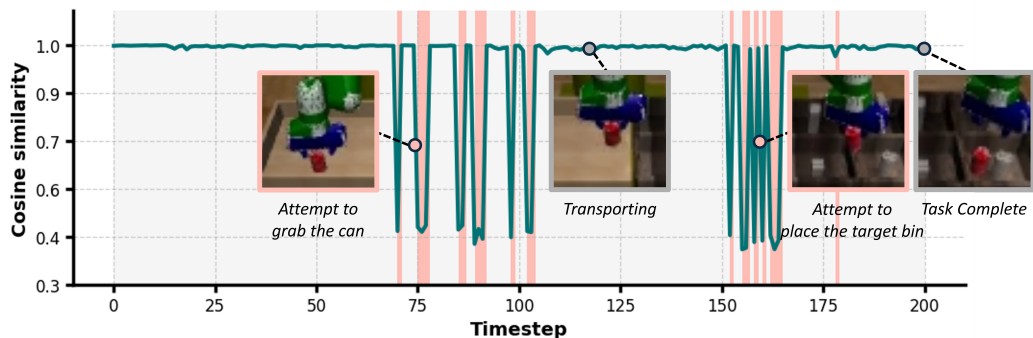

Figure 13: Similarity between actions from a previously planned chunk and newly replanned actions at each time step in the *Can* task.

## F   Performance of Additional Guidance Methods

In addition to our Self-Guidance(SG), we evaluate two additional self guidance approaches for ablation.

**Noised Observation (NO)**   Instead of utilizing past condition as negative guidance, we also try to utilized directly perturbed condition as a bad output. Specifically,

$$\hat{\epsilon}_{\text{new}} \leftarrow (1 + w) \cdot \epsilon_\theta(x, s_t) - w \cdot \epsilon_\theta(x, \, s_t + s * \delta), \quad \text{where } \delta \sim \mathcal{N}(\mathbf{0}, \mathbf{I}) \qquad (16)$$

where s denotes scailing factor. We set $s = 0.1$ empirically.

**Time-Step Guidance (TSG) [32]**   Recent work [18] introduce following Time-Step guidance. In this method, the bad output is computed by perturbed denoising timestep with same condition distribution. Specifically,

$$\hat{\epsilon}_{\text{new}} \leftarrow (1 + w) \cdot \epsilon_\theta(x, s_t, t) - w \cdot \epsilon_\theta(x, \, s_t, \tilde{t}) \qquad (17)$$

where $\tilde{t}$ denotes perturbed timestep embedding $\tilde{t} = t + s \cdot t^\alpha$ and $s, a$ are hyperparameters of TSG. We set $s = 2, \alpha = 1$, following default configuration of TSG.

**Results**   As shown in Fig. 14, 'NO' also shows worse performance than *Vanilla*. While TSG outperforms *Vanilla* slightly, its improvements is still marginal. Our SG shows remarkable performance improvement compared to other guidance methods.

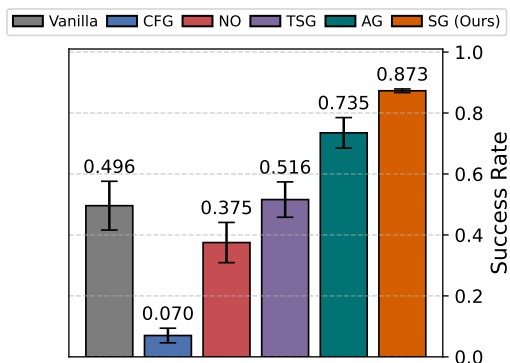

Figure 14: Compared to other guidance methods, SG achieves superior performance.

## G AutoGuidance vs. Self Guidance

To present a detailed comparison between Autoguidance (AG) and Self-Guidance (SG), we depict the performance of both methods at different guidance scales in Fig. 15. As shown, our SG clearly achieves higher optimal performance than AG across various noise scales. Moreover, while AG's optimal performance decreases rapidly as the noise scale increases, our SG maintains robust performance in noisy environments.

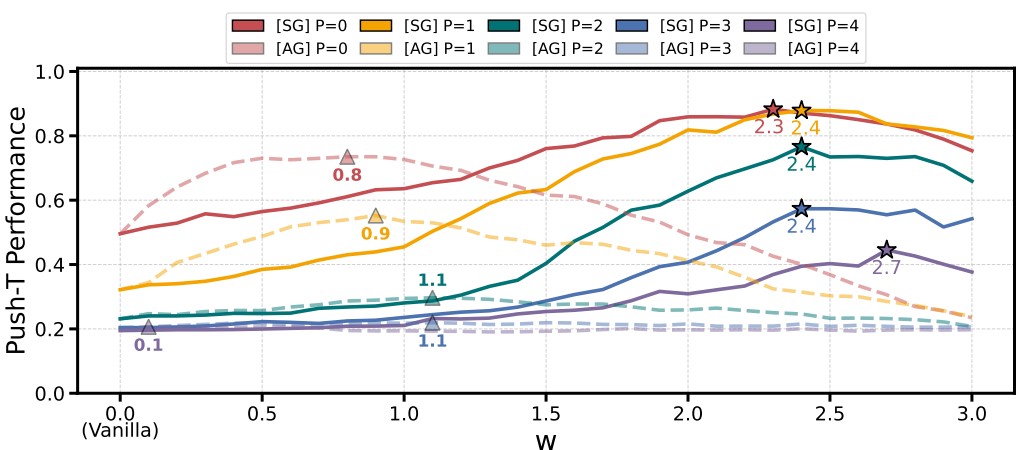

Figure 15: Effect of SG & AG guidance scale($w$) on varying noise levels (P)

## H Real World Experiments Details

This section describes the experimental details of Sec.4.2(main). We have detailed the SO-100 robot arm [28] and camera setup, training details, evaluation details for the both inference method *Vanilla DP* [9] and *Ours*.

**Experimental Setup** We perform real world experiments with SO-100 robot arm [28] with three cameras. As shown in Fig. 16, each camera records bird-eye, front, wrist view of robot arms. The input shapes of images are 1920 x 1080 px videos with 30 fps, but we down-sample the image shapes to 224 x 224 px for training and inference. To ensure consistency with the training environment, we set the robot's operation to 30 FPS during inference.

**Problem Setup** As a simplified version of Robomimic *'Can'* task [24], we consider a task that robot grasps a pen-holder grip and placing it into cup. Fig. 7(b)-(d)(main) illustrates the total sequence

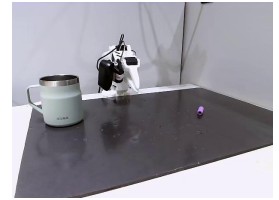 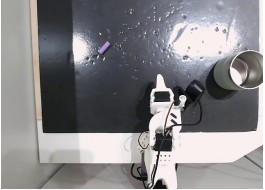 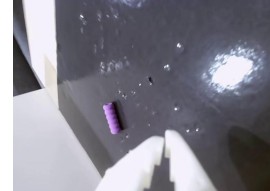

|(a) Front view | (b) Bird-Eye view | (c) Wrist view|

Figure 16: Real world robot arm camera setup. We use three 1920 x 1080 px, 30fps webcam. For training and inference, input videos are down-sampled to 224 x 224 px without cropping.

| Hyperparameter | Value |
|---|---|
| Observation history | 2 |
| Prediction horizon ($H$) | 16 |
| LR scheduler | cosine scheduler with linear warmup 500 steps |
| Learning rate | 1e-4 |
| Batch size | 32 |
| Train steps | 240K (Early stopped with max 600K steps) |
| Input image size | 224 x 224 px |
| Vision backbone | ResNet-50 |
| UNet dimension | [256,512,1024] |

Table 5: Diffusion Policy hyperparameter for real world experiments

of placing tasks. For *stochastic* scenario, we move the cup to introduce disturbance, while *static* scenario maintain the position of both pen-holder grip and cup.

**Training Details** We make 300 demonstration episodes with lerobot open source [28]. For each demonstration episodes, initial place of pen-grip holder and cup are randomly chosen while robot arm starts with same rest position. We follow the Diffusion Policy [9] training recipe with few modifications to fit in real world. We use ResNet-50 [33] as vision backbone with IMAGENET [34] pretrained weight, and cosine LR scheduler starts with linear warmup 500 steps. We use early stopped 240K steps checkpoint, which requires 27H with one NVIDIA RTX 6000 Ada Generation GPU and AMD Ryzen Threadripper PRO 7985WX CPU. Additional hyperparameter details are listed in Table. 5

**Evaluation** In Sec.4.2(main), we compare with two models, Vanilla DP [9], and Ours, using the guidance weight $w$ as 0.1 for SG, and the similarity threshold $\tau$ as 0.99 for AC. In the *static* scenario, we set four types of starting points, and measure the success rate from five experimental runs at each point, total 20 episodes. In the *stochastic* scenario, we introduce disturbance by moving the cup by hand after the robot arm grasped the pen-grip holder. We consider each evaluation episode as failure if it exceeds 30 seconds time limit or drops pen-grip holder before placing it to cup.

# I  Additional Real World Experiments

We conduct additional experiments in real-world scenarios to compare the performance of our proposed method (Ours) with that of Vanilla DP [9], EMA [22], and BID [23] in *stochastic* scenario.

**Problem Setup** Similar to the previous real world experiment with the Vanilla DP [9], we perform a task that grasp a pen-holder grip and placing it in a cup. To introduce a highly *stochastic* scenario, the cup periodically moves in a circular path. Fig. 17 visualizes the successful and failed samples of the task.

**Baselines** We collected 300 demonstration episodes of placing the pen-holder grip while the pen-holder grip and cup are in *static* scenario. The detailed hyperparameters are same to those presented in Table. 5, which is the Sec.4.2(main) experiment's baseline, excluding the batch size and the number of training steps. We trained a new baseline for 320K steps with a batch size of 16, employing a cosine warmup scheduler. For a fair comparison, we configured the baseline methods, EMA and BID, similarly to the Sect.4.1(main) experiments. For EMA, we set its decay rate $\lambda$ to 0.5. For BID, we

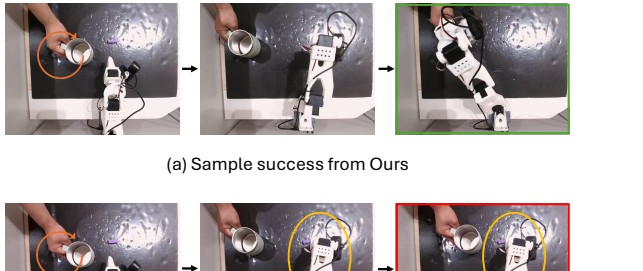

(a) Sample success from Ours

(b) Sample failure from BID

Figure 17: **Additional Real World *Stochastic* Task**: The goal is grasp a pen-grip holder and placing it to cup which periodically move along circular path. (a) Visualization of success sample from *Ours* (b) Failed sample from *BID* [23]. We observe that a few evaluation fail due to idling actions.

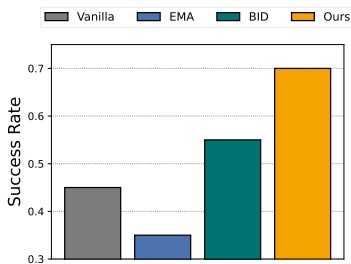

Figure 18: **Additional Real World Experiments**: We compare Vanilla DP [9], EMA [22], BID [23], and Ours under 20 *stochastic* episodes. Ours achieve 70% success rates, which is higher than other inference methods.

adopted its original settings, and choose the strong policy as 320K steps and the weak policy as 240K steps.

**Evaluation** We evaluated each method based on 20 task executions, each initiated from the same position. Fig. 18 shows the success rates of tasks in real-world experiments. EMA shows a lower success rate than the Vanilla Diffusion Policy. Both BID and Ours achieved higher success rates compared to the Vanilla DP, but Ours shows a slightly higher success rate as it performed the pen-holder gripping action more precisely. The success rate of each method was consistent with the analysis from the Sec.4.1(main) simulation experiments. As shown in Fig. 17, BID failures frequently resulted from idling actions following a grasp failure, which can also be observed in Vanilla DP and EMA. However, our method experienced fewer grasp failures and exhibited no idling actions during evaluation. We found that BID inference ran at an average of 16 Hz for each action generation, so the robot operates unsmooth and halting manner. But, Ours generate actions an average of 29 Hz, and move smoothly.

