# OpenReview forum: "Improving Generative Behavior Cloning via Self-Guidance and Adaptive Chunking"
_NeurIPS.cc/2025/Conference — NeurIPS 2025 poster_

### Official Review · Reviewer_eCfX · 2025-06-27

**Clarity:** 4
**Significance:** 3
**Originality:** 3
**Rating:** 5
**Confidence:** 4

**Summary:**

The paper presents two novel methods for improving diffusion policies: Self-Guidance (SG), which guides the diffusion process by negating the action probabilities of previous timesteps, and Adaptive Chunking (AC), which dynamically selects the length of the action chunk to balance reactivity and consistency. The experiments conducted demonstrate significant performance improvements across a variety of simulation benchmarks and real-world robotic manipulation tasks.

**Questions:**

-  In Figure 3, it is unclear from the main text what exactly "different levels of noise" refer to (diffusion policy sampling process, the environment observations, or dynamics?). Also, including a baseline result for SG at w=0 (vanilla) would significantly improve interpretability, clearly illustrating the gains specifically attributable to SG.

- The authors use cosine similarity to measure action similarity. However, this metric is primarily suitable for position control tasks. It should be clarified that for tasks involving joint angles or velocity control, metrics like L1 or L2 norms might be more appropriate.

- In the experimental section, it is stated that closed-loop control was used for stochastic environments. This choice seems contradictory since the key advantage of adaptive chunking is precisely to automatically decide on the execution length. Either this choice needs clarification, or the experiment should be restructured to genuinely test AC.

- Regarding the VLA experiment in the discussion, can something like that be done for ACT? I'm aware that such an experiment is out of the scope of a rebuttal, just interested in your thoughts on the matter.

**Ethical Concerns:**

["NO or VERY MINOR ethics concerns only"]

**Final Justification:**

The authors addressed all the points I raised in my rebuttal. One of the weaknesses was an error in their derivation, which they fixed, and for the other two, they provided sufficient explanation. I don't see any clear weakness in their results and think that this paper should be accepted.

**Limitations:**

yes

**Paper Formatting Concerns:**

All is well

**Quality:**

4

**Strengths And Weaknesses:**

**Strengths:**

- The proposed methods (SG and AC) are innovative, simple, and practically appealing, requiring minimal changes to existing diffusion policy frameworks.

- The experiments are comprehensive, covering both simulation benchmarks and real-world scenarios, with clear and statistically significant performance improvements.

- The paper is written in a clear and easy-to-follow way.

**Weaknesses:**

- The paper could provide a more thorough intuitive discussion about cases where SG might not be beneficial. For example, in slow-changing scenarios (e.g., a robot continuing to move toward the same target for some time), the SG method seems likely to reduce the probability of repeatedly sampling the optimal action.

- Equations 12 and 13 contain an error. The approximation for first-order differences is not as mentioned, but $f(x+\delta x) = 2f(x)-f(x-\delta x)$. This also invalidates the discussion following the equation.

- While the authors argue that adaptive chunking can mitigate issues due to unexpected changes in the environment, it is less clear how it prevents jittering arising from sampling multimodal trajectories. For instance, consider a scenario where a robot must pass an obstacle, which can be approached either from the left or from the right. If the current action chunk in the queue directs the robot to move left, but the next sampled action chunk directs it to move right, adaptive chunking would replace the current action. In subsequent steps, the sampled action could revert again to the left, causing continuous toggling between these trajectories rather than smooth behavior. Clarifying or experimentally addressing this concern would strengthen the paper. Clarifying or experimentally addressing this concern would strengthen the paper.

---

> ### Author Rebuttal · Authors · 2025-07-29
>
> We are grateful for your insightful feedback and for acknowledging the contributions of our work, especially highlighting our method's innovation, simplicity, and statistically significant performance improvements. We have carefully considered each of your points and present our detailed responses below.
>
> ## Weakness 1
> Thank you for the insightful suggestion. However, **our SG is not negatively impacted by slowly changing scenarios**. In these situations, where the change in state $|s_t - s_{t-\delta t}|$ is small, the ratio $\frac{p_\theta(a_t|s_t)}{p_\theta(a_t|s_{t-\delta t})}$ approaches 1. Consequently, Eq. (11) simplifies to $p(a_t|s_t)$. Thus, our SG is not adversely affected by slowly changing scenarios unless an extremely large guidance weight, $w$, is used.
>
> Conversely, the scenario where SG is less beneficial is when the robot move very quickly. This rapid movement creates a large observation gap, $|s_t - s_{t-1}|$, which in turn causes SG to produce a faster (larger) action in the subsequent step due to large guidance signal. This can create a positive feedback loop, leading to progressively larger observation gaps. However, this issue can be mitigated in practice by filtering out excessively fast actions using a thresholding technique, mechanisms that are commonly built into robotic systems due to inherent motor speed limitations. We will add this discussion in final version.
>
>
> ## Weakness 2
> Thank you for pointing this out. This was a **typographical error**, we mistakenly used ‘+’ instead of ‘–’. However, the subsequent discussion remains valid despite this mistake. The correct version of Eq. (12) is:
>
> $$
> (1 - w)\cdot \epsilon(x, s_t) + w\cdot (2\cdot \epsilon(x, s_t) - \epsilon(x, s_{t - \Delta t}))
> $$
>
> We will correct this in the final version of the paper.
>
> ## Weakness 3
> Thank you for your valuable feedback. The toggling scenario you described rarely occurs in practice and is, in fact, **prevented by our proposed methodology** for the following reasons:
>
> 1.  At a specific point during the rollout, the policy model $\pi(a|s_t)$ outputs a multi-modal distribution over actions. The robot samples an action from one mode according to its probability and executes the corresponding control.
>
> 2.  Once the robot has committed to one mode, the model predicts the subsequent action, $\pi(a|s_{t+1})$. However, because the robot has already entered a specific mode, the probability of the model sampling an action corresponding to the opposite mode is significantly lower than it was at the previous timestep. Consequently, toggling between modes is unlikely, even with Adaptive Chunking alone.
>
> 3.  Furthermore, applying our **Self-Guidance** (as shown in Eq. (11)) resolves this problem as well. Since the probability of the alternative mode has already decreased, Self-Guidance further **attenuates the probability of the opposite mode** while reinforcing the probability of the currently selected one.
>
> 4.  For instance, consider a simplified two-mode scenario (e.g., left/right). Assume the robot has just enter one mode, and the probabilities evolve as follows: the previous probability was $p(a|s_{t-1}) = [0.5, 0.5]$, and the (unguided) current probability is $p(a|s_{t}) = [0.7, 0.3]$. With a guidance weight of $w=2$, our method updates the probability based on Eq. (11). The new guided probability, $p_{SG}(a|s_{t})$, is proportional to:
>     $$p_{SG}(a|s_{t}) \propto p(a|s_{t}) \left( \frac{p(a|s_{t})}{p(a|s_{t-1})} \right)^w$$
>     Calculating this element-wise:
>     $$p_{SG} \propto [0.7, 0.3] \odot \left( \frac{[0.7, 0.3]}{[0.5, 0.5]} \right)^2$$
>     $$p_{SG} \propto [0.7, 0.3] \odot [1.96, 0.36] = [1.372, 0.108]$$
>     After normalization, the final probability becomes $[0.927, 0.073]$. As shown, the probability of selecting the other mode is significantly attenuated ($0.3 \rightarrow 0.073 $ ).
>
> While this is a simplified example, we observe in our experiments that the model prevents toggling for the reasons described above. This analysis underscore the importance of using both of our methods, SG and AC. We will add a more detailed explanation and relevant visualizations of this situation to the final version of the paper.
>
> Additionally, the "jittering problem," as we define it in our paper, refers to the 'jerky' or 'trembling' movements that occur while the robot follows a trajectory from a specific mode, due to stochastic nature of the generative model. This problem is well discussed in recent work [1,2] and is known to negatively impact overall rollout performance, particularly in real-world scenarios. The primary objective of our Adaptive Chunking is to mitigate this issue.
>
> ## Question 1
> - **Figure 3 P :** Please see Appendix A. **we denotes $P$ as an environmental noise**, which is the speed of movement of T block and is the same noise setting in previous works, BID \[1].  We will clarify in final version.
> - **SG at w=0 :** **Please see Figure 5 in Appendix**. As shown in figure, our SG largely surpass the $w=0$ (vanilla) with large margin and even surpass the more computationally heavy technique, AG (AutoGuidance) \[6] in all settings. We will append this result in our Figure 3 (main). Thanks for your constructive feedback.
> - **Remark** : Moreover, we'd like to remark again that this is the **first work that enhances the performance of diffusion-based behavior cloning by using a guidance technique**, as far as we know. While guidance techniques, especially CFG, are a de facto standard for improving performance in image and video generation, they have remained largely unexplored in the domain of **action generation** because of their poor performance in BC (See Sec. 3.3 of [5]). In contrast, we are the first to demonstrate that guidance techniques can be applied to diffusion-based behavior cloning by using extrapolation-based techniques, such as AG or our SG. Furthermore, we reveal that the mechanism for this improvement is the elimination of low-fidelity actions. **We believe this finding is a significant contribution that can be immediately applied to the rapidly advancing field of VLA community.**
>
> ## Question 2
> Thank you for your insightful feedback. As shown in Table 2 in the Appendix, we found that the cosine-similarity-based method outperforms other metrics in all benchmarks we tested, which represent a wide range of standard manipulation tasks. However, we also agree that for some action types (e.g., velocity-based), other metrics can be more useful. We will clarify this point in the final version.
>
> ## Question 3
> Thanks for the feedback. The term 'closed-loop' for Adaptive Chunking means that its update rule (Eq. (14)) is applied at every timestep. Conversely, the 'open-loop' version, which we used in the static environment for a fair comparison with the baselines, applies the update rule only every 8 timesteps. You can also check the `codes/improvingGBC/diffusion_policy/sampler/single.py:25` for more specific implementation. The current Eq. (14) denotes this closed-loop implementation. We agree that this can be misleading and will clarify it in the final version.
>
> ## Question 4
> **Yes**. While our paper mainly focused on Diffusion Policy , we believe our method can be applied to any behavioral cloning framework as long as it use  action chunking and use probabilistic framework to model action space.
>
> For instance, in the case of ACT [2], which uses a VAE decoder to model action probability, we can simply apply Self-Guidance by extrapolating the input to the VAE decoder (or the output of the transformer decoder) using the value from the previous timestep. Adaptive Chunking could also be seamlessly integrated into the ACT framework, similar to their temporal ensemble technique.
>
> Moreover, to validate our method's wide applicability claim, we present results from applying our method to other state-of-the-art diffusion-based VLAs that use open-loop control: $\pi_0$ [3] and GR00T-N1 [4].   Specifically, we experiment the $\pi_0$ model in both a real-world setup and LIBERO simulation benchmark, and GR00T-N1 in real world setup.  We employ same setting of Sec. 4.2 (of our main paper) for real world experiment and Sec.5 (of our main paper) for LIBERO simulation evaluation.
>
>
> ### $\pi_0$
>
> | (Real) | **Ours** | $\pi_0$ (open loop; vanilla) | $\pi_0$ (closed loop) |
> | :--- | :---: | :---: | :---: |
> | Static | **85%** | 75% | 70% |
> | Stochastic | **65%** | 45% | 50% |
>
> | LIBERO (simul) | P=1 | P=3 | P=5 |
> | :--- | :---: | :---: | :---: |
> | $\pi_0$ (open loop; vanilla) | 0.786 | 0.424 | 0.094 |
> | $\pi_0$ (closed loop) | 0.786 | 0.486 | 0.208 |
> |  **Ours** | **0.828** | **0.526** | **0.222** |
>
> ### GR00T-N1
>
> | (Real) | Ours | GR00T-N1 (open loop; vanilla) | GR00T-N1 (closed loop) |
> | :--- | :---: | :---: | :---: |
> | Static | **85%** | 75% | 25% |
> | Stochastic | **45%** | 20% | 20% |
>
>
> As shown, when applied to $\pi_0$ and GR00T-N1,  **our method achieves the best results in all settings**. While their vanilla (open-loop) suffers a drastic performance decrease due to stochastic noise and the closed-loop variant shows only marginal improvement, our method demonstrates a consistent and significant improvement in every setting. This underscores our method's wide applicability and its potential to other VLAs, such as ACT.
>
>
> -- references --
> [1] Liu, Yuejiang, et al. "Bidirectional decoding: Improving action chunking via closed-loop resampling.", ICLR 2025
> [2] Zhao, Tony Z., et al. "Learning fine-grained bimanual manipulation with low-cost hardware."
> [3] Black, Kevin, et al. "$\pi_0 $: A Vision-Language-Action Flow Model for General Robot Control."
> [4] Bjorck, Johan, et al. "Gr00t n1: An open foundation model for generalist humanoid robots." arXiv preprint arXiv:2503.14734 (2025).
> [5] Pearce, Tim, et al. "Imitating human behaviour with diffusion models." , ICLR 2023.
> [6] Karras, Tero, et al. "Guiding a diffusion model with a bad version of itself." NeurIPS 2024

---

> > ### Comment · Reviewer_eCfX · 2025-08-02
> >
> > I thank the authors for their responses. Especially for their response to “weakness 3”, which shows the advantage of combining both SG and AC, where SG guardrail against a potential problem in AC. I believe some version of this explanation can contribute to the paper.
> > After reading the responses and going over the paper once again, I will raise my score.

---

### Official Review · Reviewer_tuvk · 2025-06-29

**Clarity:** 3
**Significance:** 2
**Originality:** 3
**Rating:** 4
**Confidence:** 4

**Summary:**

This paper presents two training-free inference modules, Self-Guidance and Adaptive Chunking, to tackle low-fidelity action sampling and balance reactivity with temporal consistency in diffusion-based behavior cloning. Self-Guidance sharpens the action distribution by using the model’s past state predictions as negative guidance during denoising, thereby biasing sampling toward higher-fidelity and forward-looking actions without extra training. Adaptive Chunking dynamically alternates between open-loop and closed-loop control based on cosine similarity of successive actions, preserving consistency in bulk motions while enabling rapid replanning during precision phases. This work yields substantial performance gains over vanilla Diffusion Policy and state-of-the-art BID method in both simulation and real-robot tests.

**Questions:**

1. Could you clarify and empirically demonstrate your contributions different from AutoGuidance and BID? For example, are there task settings where those baselines fail but your methods succeed?
2. Please extend OpenVLA-OFT evaluations to include direct comparisons with diffusion-head VLA architectures such as $\pi_0$, UniVLA, or similar methods on standard benchmarks (e.g., LIBERO). This will show whether your inference modules work for state-of-the-art policies instend of pure diffusion policies only.
3. I would expect more real-world experiements on extension to other VLAs.

**Ethical Concerns:**

["NO or VERY MINOR ethics concerns only"]

**Final Justification:**

The authors have addressed most of my concerns except the contribution about introducing two plug-and-play inference techniques is a little fair. But they show superior performance and the quality of this paper is above the borderline. So, I suggest borderline accept.

**Limitations:**

Yes

**Quality:**

3

**Strengths And Weaknesses:**

**Strengths**

The proposed methods are plug-and-play and computationally efficient, requiring no retraining or architectural changes, yet consistently improve action fidelity and responsiveness across diverse manipulation tasks.

**Weakness**

1. The contribution seems limited: While the methods improve existing diffusion policies, they mainly adapt known ideas from classifier-free and auto-guidance. The novelty over AutoGuidance [1] and prior chunking strategies (i.e., BID) is incremental and could be more sharply distinguished.

2. The extension to VLA-based models (OpenVLA-OFT) is still at an early stage and is not benchmarked against recent diffusion-head VLA architectures such as $\pi _0$ or other $\pi _0$-style methods (like UniVLA), leaving open the question of these techniques' performance on state-of-the-art policies. I would expect more real-world experiements on extension to other VLAs.

3. The real-world robot is a little toy, but it's understandable.

**References**
[1] Karras, Tero, et al. "Guiding a diffusion model with a bad version of itself." Advances in Neural Information Processing Systems 37 (2024): 52996-53021.

---

> ### Author Rebuttal · Authors · 2025-07-29
>
> Thank you for your insightful feedback and for highlighting our method's simplicity and computational efficiency, which can be directly applied to many generative model-based control systems without incurring cost. Below, we have carefully considered each of your points and present our detailed responses.
>
>
> ## Weakness 1, Question 1
> > The contribution seems limited: While the methods improve existing diffusion policies, they mainly adapt known ideas from classifier-free and auto-guidance. The novelty over AutoGuidance [1] and prior chunking strategies (i.e., BID) is incremental and could be more sharply distinguished.
>
> > Could you clarify and empirically demonstrate your contributions different from AutoGuidance and BID?
>
> We'd like to remark again on our contribution and novelty against AG and BID as follows:
>
> -   **Guidance**: As far as we know, this is the **first work that enhances the performance of diffusion-based behavior cloning by using a guidance technique**. While guidance techniques, especially CFG, are a de facto standard for improving performance in image and video generation, they have remained largely unexplored in the domain of **action generation**. A key reason for this is that CFG performs poorly for action generation (See Sec. 3.3 of [1] and the table below), or it requires training a separate goal-state classifier [4], which lacks generalizability. In contrast, we are the first to demonstrate that guidance techniques can be applied to diffusion-based behavior cloning by using extrapolation-based techniques, such as AG or our SG. Furthermore, we reveal that the mechanism for this improvement is the elimination of low-fidelity actions. **We believe this finding is a significant contribution that can be immediately applied to the rapidly advancing field of VLA community.**
>
> ---
>
> -   **Superior Performance**: As shown in Fig. 3 and Fig. 5 of the main paper and Fig. 1,  Fig. 4, Fig. 5 of the Appendix, our approach (Ours/SG+AC) consistently outperforms both AG and BID across a wide range of settings, including various datasets, hyperparameters, noise levels, and both simulation and real-world environments. **These results suggest that our method is not limited to improving performance in specific scenarios but is rather a robust and statistically significant improvement that generally enhances performance**, also highlighted as a key strength by Reviewer eCfX.
>     -   Moreover, to further validate our claim, we conducted an ablation study with results for BID, CFG, AG, and each guidance method augmented with our Adaptive Chunking (AC). As the table clearly indicates, our SG+AC method achieves the highest average performance, reinforcing the robustness and superiority of our approach.
>
>         |             | Vanilla | Ours (SG+AC) | BID (ICLR'25) | CFG   | CFG+AC | AG    | AG+AC |
>         | :---------- | :------ | :---------- | :----------- | :---- | :----- | :---- | :---- |
>         | **Push-T** | 0.496   | 0.819       | 0.652        | 0.071 | 0.473  | 0.722 | 0.826 |
>         | **Square**| 0.633   | 0.753       | 0.653        | 0.000 | 0.000  | 0.497 | 0.607 |
>         | **Lift** | 0.157   | 0.650       | 0.447        | 0.020 | 0.017  | 0.340 | 0.843 |
>         | **Can** | 0.787   | 0.927       | 0.813        | 0.000 | 0.000  | 0.667 | 0.867 |
>         | **Kitchen** | 0.298   | 0.412       | 0.410        | 0.137 | 0.138  | 0.294 | 0.457 |
>         | **Transport**| 0.387   | 0.587       | 0.437        | 0.000 | 0.000  | 0.310 | 0.350 |
>         | **Average** | 0.460   | **0.691** | 0.569        | 0.038 | 0.105  | 0.471 | 0.658 |
>
> ---
>
> - **Computational cost :** While Autoguidance and BID incur significant computational costs, our method effectively resolves these issues. Specifically, Autoguidance and BID have the following problems in terms of computational costs:
>
>   -   **v.s. Autoguidance (AG):**
>       -   ***Storage Overhead***: AG introduces new hyperparameters for selecting a "weak" model, which necessitates saving model weights at every epoch during training. This leads to substantial storage overhead.
>       -   ***Memory Cost***: AG **doubles the memory cost** (especially VRAM) by requiring weights for both strong and weak models. This increased memory overhead is particularly problematic for deployment on resource-constrained edge devices.
>       -   ***Latency***: AG doubles the computational cost for inference. Unlike our SG, which can leverage batched computation efficiently, AG's use of different weights on the same input prevents effective batching and effectively **doubles the practical latency**, which is a critical issue for real-time applications.
>       -   In contrast, our Self-Guidance (SG) resolves all these computational issues, **having 2x faster latency while achieving higher performance than AG**.
>
>   -   **v.s. BID:**   The BID "forward contrast" process requires generating `N` (usually 16) samples from both the strong and weak models. Hence, it suffers from all the aforementioned drawbacks of Autoguidance  and also multiplies the computational cost by a factor of `N`, which is significantly large computational overhead, especially for edge devices. Our method resolves all these issues, **having 16x smaller computational cost and 2x faster latency while simultaneously delivering superior performance to BID.**
>
> We will add clarification of our novelty claims and comparison of AG and BID in the final version of paper.
> ## Weakness 2,3, Question 2,3
>
> > The extension to VLA-based models is not benchmarked against recent diffusion-head VLA architectures such as $\pi_0$ ... I would expect more real-world experiments on extension to other VLAs.
>
> > Please extend OpenVLA-OFT evaluations to include direct comparisons with diffusion-head VLA architectures such as $\pi_0$ on standard benchmarks (e.g., LIBERO).
>
> > The real-world robot is a little toy, but it's understandable. I would expect more real-world experiments on extension to other VLAs.
>
> Thanks for the suggestion. We conducted additional experiments on other diffusion based state-of-the-art VLAs that use open loop control, **$\pi_0$** [2] and **GR00T-N1** [3].  Specifically, we experiment the $\pi_0$ model in both a real-world setup and LIBERO simulation benchmark, and **GR00T-N1** in real world setup.  We employ the same setting of Sec. 4.2 (of our main paper) for real world experiment and Sec.5 (of our main paper) for LIBERO simulation evaluation.
>
>
> ### $\pi_0$
>
> | (Real) | **Ours** | $\pi_0$ (open loop; vanilla) | $\pi_0$ (closed loop) |
> | :--- | :---: | :---: | :---: |
> | Static | **85%** | 75% | 70% |
> | Stochastic | **65%** | 45% | 50% |
>
> | LIBERO (simul) | P=1 | P=3 | P=5 |
> | :--- | :---: | :---: | :---: |
> | $\pi_0$ (open loop; vanilla) | 0.786 | 0.424 | 0.094 |
> | $\pi_0$ (closed loop) | 0.786 | 0.486 | 0.208 |
> |  **Ours** | **0.828** | **0.526** | **0.222** |
>
> ### GR00T-N1
>
> | (Real) | Ours | GR00T-N1 (open loop; vanilla) | GR00T-N1 (closed loop) |
> | :--- | :---: | :---: | :---: |
> | Static | **85%** | 75% | 25% |
> | Stochastic | **45%** | 20% | 20% |
>
> As shown, when applied to $\pi_0$ and GR00T-N1,  **our method achieves the best results in all settings**. While their vanilla (open-loop) suffers a drastic performance decrease due to stochastic noise and the closed-loop variant shows only marginal improvement, our method demonstrates a consistent and significant improvement in every setting. This underscores our method's wide applicability and its potential for enhancing future VLA-style models. We will include these additional VLA experimental results in the final version.
>
>
> --- references ---
>
> [1] Pearce, Tim, et al. "Imitating human behaviour with diffusion models." , ICLR 2023.
> [2] Black, Kevin, et al. " $\pi_0$ : A Vision-Language-Action Flow Model for General Robot Control."
> [3] Bjorck, Johan, et al. "Gr00t n1: An open foundation model for generalist humanoid robots." arXiv preprint arXiv:2503.14734 (2025).
> [4] Janner, Michael, et al. "Planning with diffusion for flexible behavior synthesis." arXiv preprint arXiv:2205.09991 (2022).

---

> ### Comment · Reviewer_tuvk · 2025-08-03
>
> Thanks for the authors' rebuttal and the new empirical results—those additional comparisons on $\pi_0$ and GR00T-N1 help strengthen the claim about wide applicability.
>
> But, I **don’t agree** with the absolute phrasing that this is “the first work that enhances the performance of diffusion-based behavior cloning by using a guidance technique.” Prior work (e.g., AdaptDiffuser [1]) has already shown that *guidance—specifically reward-gradient-based guidance to generate and select high-quality synthetic expert data—can be leveraged to improve diffusion-based planning/behavior cloning performance*. A more precise and defensible claim would be that this is the first work to apply the particular *self-guidance* mechanism (using past state predictions as negative guidance to eliminate low-fidelity actions) in diffusion-based behavior cloning, and to pair that with the proposed adaptive chunking scheme for balancing reactivity and temporal consistency.
>
> **All in all, the paper introduces two practical, plug-and-play inference techniques that yield consistent gains; my remaining concern is that the core conceptual novelty is somewhat mild.**
>
> [1] Liang, et al. "AdaptDiffuser: diffusion models as adaptive self-evolving planners." ICML. 2023.

---

> > ### Author Response · Authors · 2025-08-04
> > **Response to Reviewer tuvk**
> >
> > > Prior work (e.g., AdaptDiffuser) has already shown that guidance can be leveraged to improve diffusion-based planning ...
> >
> > Thank you for the engaging discussion and constructive feedback. We acknowledge that prior works like AdaptDiffuser and [1,2] also use guidance, and we will surely add these references to enrich discussion.
> >
> > To be precise, our claim is that this is the first guidance technique to improve performance in **'Goal-Agnostic Behavior Cloning (GABC)'** tasks, which is the standard paradigm for modern visuomotor control [5] and VLAs. The AdaptDiffuser and prior works [1,2] fall under a different category, **'Goal-Conditioned Imitation Learning (GCIL)'**, which requires an explicit, task-specific definition of a 'goal state' (e.g., the exit of a maze) or an explicit reward function. In this setting, common diffusion guidance techniques like Classifier Guidance and CFG can be easily applied due to the explicit optimality condition, but these methods often lack the generalization required for complex, open-ended tasks.
> >
> >
> > In contrast, our **GBC tasks**, or **GABC**, relies only on an expert's demonstration `(state, action)` pairs, enabling the general, low-level control that is standard in modern visuomotor and VLA systems. However, this formulation makes it impossible to leverage existing guidance techniques for the following reasons:
> >
> > * **(a) Classifier-Guidance:** The absence of an explicit goal or reward function makes it impossible to train the required classifier.
> > * **(b) Classifier-Free Guidance (CFG):** The implicit reward of CFG in BC, `p(state|action)`, is not a meaningful reward for control tasks [3], in contrast to `p(class|data)`, which is successful in image generation [4].
> >
> > For these reasons, while it is well-known that the performance of diffusion models heavily depends on guidance, its application to GBC has remained largely unexplored. This has created a significant gap between recent research on diffusion guidance and action generation, hindering the huge potential of diffusion models in control.
> >
> > ***
> >
> > **Our work is the first to apply general guidance to GABC and fill this gap.** We show that our novel **training-free, plug-and-play** self-guidance can significantly improve control performance in a goal-agnostic setting, especially from simple diffusion policy to complex modern VLAs. Moreover, we not only show empirical success, but also provide an in-depth analysis that this guidance promotes higher reactivity, filters low-fidelity actions, and implicitly considers future states. **We still believe this is a novel and significant contribution.**
> > We will clarify this in the final version.
> >
> > > ... my remaining concern is that the core conceptual novelty is somewhat mild.
> >
> > In addition to the above contribution, we'd like to further clarify our novelty against AutoGuidance and BID.
> >
> > * **v.s Autoguidance Novelty**
> >
> > Looking at recent research trends in vision generation, a variety of **Self-Guidance** techniques have been actively proposed to solve the aforementioned problems of Autoguidance, such as PAG [6, ECCV'24], TSG [7, ICLR'25], and STG [8, CVPR'25]. We believe that this trend suggests that **developing efficient, plug-and-play, and high-performing self-guidance techniques is recognized as a novel and significant research contribution in itself.** We develop a novel self-guidance from a **control-centric perspective** and show general performance improvement. Moreover, we also show that in control tasks, our SG is more effective than these SOTA self-guidance methods, as shown in Fig. 4 of the Appendix.
> >
> > * **v.s BID Novelty**
> >
> > As previously mentioned, BID's primary drawback is its computational cost, requiring 16x more computation and resulting in 2x slower latency than our method. Our method resolves this while also outperforming BID in almost all settings. **We argue that developing a method that is both more efficient and more effective than the previous SOTA is a significant novelty and contribution in itself.**
> >
> >
> > Considering these points, we kindly ask you to reconsider the novelty of our work. We thank you once again for your time and insightful review, and we welcome any further discussion.
> >
> >
> > --- references ---
> > [1] Janner et al. "Planning with diffusion for flexible behavior synthesis." ICML. 2022.
> > [2] Dong et al. "Diffuserlite: Towards real-time diffusion planning." NeurIPS. 2024.
> > [3] Pearce et al. "Imitating human behaviour with diffusion models." ICLR. 2023.
> > [4] Ho et al. "Classifier-free diffusion guidance." arXiv. 2022.
> > [5] Chi et al. "Diffusion policy: Visuomotor policy learning via action diffusion." RSS. 2023.
> > [6] Ahn et al. "Self-rectifying diffusion sampling with perturbed-attention guidance." ECCV. 2024.
> > [7] Sadat et al. " Rethinking classifier-free guidance for diffusion models." ICLR. 2025.
> > [8] Hyung et al. "Spatiotemporal skip guidance for enhanced video diffusion sampling." CVPR. 2025.

---

> > > ### Comment · Reviewer_tuvk · 2025-08-04
> > >
> > > Thanks for you detailed reply. I think the paper is above the borderline and I would raise my score to 4 (borderline accept) althogh I still think the paper's significance is ordinary.

---

### Official Review · Reviewer_fbxt · 2025-07-02

**Clarity:** 3
**Significance:** 2
**Originality:** 2
**Rating:** 5
**Confidence:** 4

**Summary:**

This paper proposes an improved generative behavior cloning (GBC) framework, specifically targeting the application of diffusion strategies in robot learning. The authors point out two main problems of current diffusion strategies in open - loop (OL) control: (1) Randomness in the diffusion process may lead to incorrect action sampling; (2) OL control lacks the ability to quickly respond to dynamic environments. To address these issues, the paper proposes two innovative methods: Self - Guidance (SG) and Adaptive Chunking (AC).

**Questions:**

1. The self-guidance (SG) formula (Eq. 10) uses past states $ s_{t-\Delta t} $ as negative guidance.
  - How was the time window $ \Delta t $ optimized? Was it task-specific or theoretically derived?
  - Were ablation studies conducted to compare SG with explicit future-state prediction?
2. Why not use probabilistic metrics (e.g., KL divergence) for uncertainty-aware transitions?
3. Does fixed-threshold chunking fail in edge cases (e.g., sudden disturbances)?
4. Would SG’s reactivity degrade with deformable objects or non-stationary tasks?

**Ethical Concerns:**

["NO or VERY MINOR ethics concerns only"]

**Final Justification:**

Your comprehensive replies fully address our technical concerns. I will raise my score to 5.

**Limitations:**

yes

**Quality:**

2

**Strengths And Weaknesses:**

Strengths:
1. Proposes two innovative methods: Self - Guidance (SG) and Adaptive Chunking (AC).
2. Adapts SG to OpenVLA , showing applicability beyond diffusion models.
3.  Empirical validation with real-world experiments.
Weaknesses:
1. AC’s step-wise similarity evaluations incur latency comparable to closed-loop control.
2. Experimental design exhibits ​limited breadth in baseline comparisons. Boader comparisons would better contextualize its advancements.
3. SG’s extrapolation (Eq. 13) assumes local smoothness of ϵ, which may not hold for highly non-linear dynamics.

---

> ### Author Rebuttal · Authors · 2025-07-29
>
> Thank you very much for your positive feedback and constructive suggestions. We have carefully considered each of your points and present our detailed responses below.
>
> ## Weakness 1
> > AC’s step-wise similarity evaluations incur latency comparable to closed-loop control.
>
> Thank you for your valuable feedback. Like prior work \[1], our approach assumes that **modern GPUs are capable of supporting the computational demands of closed-loop control without compromising the standard visuomotor control frequency (\~30Hz).** In practice, our real-world experiments with a 10-step diffusion policy on an A6000 GPU comfortably maintains the ~30Hz control rates requirement.
>
> **We would like to clarify that the primary contribution of this work lies in maximizing control accuracy under this assumption**, which we have validated through extensive experimental results.
>
>
>
> ## Weakness 2
> > Experimental design exhibits ​limited breadth in baseline comparisons. Boarder comparisons would better contextualize its advancements.
>
> Thank you for your feedback. To provide a broader comparison with additional baselines, we conducted further experiments on two state-of-the-art diffusion-based VLAs that use open-loop control: **\$\pi\_0\$** \[2] and **GR00T-N1** \[3]. Specifically, we evaluated **\$\pi\_0\$** in both real-world settings and the LIBERO simulation benchmark, and **GR00T-N1** in the real-world setting. For consistency, we followed the same experimental setup as described in Section 4.2 of the main paper for real-world evaluation, and Section 5 for the LIBERO benchmark.
>
> ### $\pi_0$
>
> | (Real) | **Ours** | $\pi_0$ (open loop; vanilla) | $\pi_0$ (closed loop) |
> | :--- | :---: | :---: | :---: |
> | Static | **85%** | 75% | 70% |
> | Stochastic | **65%** | 45% | 50% |
>
> | LIBERO (simul) | P=1 | P=3 | P=5 |
> | :--- | :---: | :---: | :---: |
> | $\pi_0$ (open loop; vanilla) | 0.786 | 0.424 | 0.094 |
> | $\pi_0$ (closed loop) | 0.786 | 0.486 | 0.208 |
> |  **Ours** | **0.828** | **0.526** | **0.222** |
>
> ### GR00T-N1
>
> | (Real) | Ours | GR00T-N1 (open loop; vanilla) | GR00T-N1 (closed loop) |
> | :--- | :---: | :---: | :---: |
> | Static | **85%** | 75% | 25% |
> | Stochastic | **45%** | 20% | 20% |
>
> As shown, **our method achieves the best performance across all settings when applied to both \$\pi\_0\$ and GR00T-N1**. While their vanilla (open-loop) versions suffer from significant performance degradation due to stochastic noise, and their closed-loop variants offer only marginal gains, our approach consistently delivers substantial improvements in every scenario. This highlights both the broad applicability of our method and its potential to enhance future VLA-style models.
>
> Thank you again for the valuable suggestion; these additional results further strengthen our claims. We will include them in the final version of the paper.
>
>
> ## Weakness 3 ,  Question 4.
> > SG’s extrapolation (Eq. 13) assumes local smoothness of ϵ, which may not hold for highly non-linear dynamics.
>
> > Would SG’s reactivity degrade with deformable objects or non-stationary tasks?
>
> Thank you for your valuable feedback and question. Our control system typically operates at around **30Hz**, meaning the time interval, $\Delta t$, between the current and past state is only about **0.033 seconds**. We posit that this short duration is sufficient for the **system's dynamics to be reasonably approximated as linear**, also numerically validated at answer of ***Question 1***.
>
> However, we acknowledge the reviewer's point that this assumption of local linearity for our Self-Guidance (SG) may not hold in environments with extremely non-linear dynamics, such as highly non-stationary dynamics or deformable objects you mentioned,  even with such a short time interval.
>
> The primary objective of our research, though, is to enhance performance on established benchmarks that closely resemble real-world scenarios where this assumption generally holds, such as the well-established Robomimic or LIBERO benchmarks. As demonstrated in our work, our proposed SG significantly improves control performance in such benchmarks, providing strong empirical evidence that our assumption is effective and practical for its intended applications.
>
> ## Question 1
> > How was the time window  optimized? Was it task-specific or theoretically derived?
>
> * **Time Window:** In all experiments presented in the paper, we use the immediate past state, **\$s\_{t-1}\$**, as input. This choice is supported both empirically, yielding the highest control performance, and theoretically, based on the intuition that the most recent state offers the most accurate and relevant information for predicting future actions. As discussed in the paper, SG can be viewed as an extrapolation of a continuous action signal using discretized differences. From this perspective, a narrow time window is essential for accurately estimating future trajectories.
>
> > Were ablation studies conducted to compare SG with explicit future-state prediction?
>
> * **Explicit Comparison:** In response to your question, we evaluate the precision of the predicted actions by computing the $L_2$ distance between the future actions generated by SG and the ground-truth actions in the Push-T environment. Specifically, we experiment with the formulation
>
> $$
> \epsilon_{\text{SG}} \leftarrow (1 + w)\epsilon_{\theta}(x \mid s_t) - w\epsilon_{\theta}(x \mid s_{t - k})
> $$
>
> for various values of $k$, to examine how incorporating past states influences prediction accuracy.
>
>
>   | | k=1 | k=2 | k=3 | k=4 | k=5 | baseline ($a_t - a_{t-1}$) |
>   |---|---|---|---|---|---|---|
>   | $L_2$ | **8.164** | 9.616 | 10.757 | 11.389 | 12.338 | 10.667 |
>
>   we denote baseline ||$a_t - a_{t-1}$|| to represent the natural variance between consecutive ground-truth actions.
>
>   As the results show, our SG with $k=1$ achieves the lowest $L_2$ error (8.164), predicting the future action more accurately than the baseline (10.667). This provides strong empirical evidence that our extrapolation mechanism operates effectively under the locally linear dynamics. We will add this experiment to the final version of the paper.
>
> ## Question 2
> > Why not use probabilistic metrics (e.g., KL divergence) for uncertainty-aware transitions?
>
> Computing exact probabilities of generated samples for divergence-based metrics incurs significant computational overhead. For instance, Kernel Density Estimation (KDE) \[4], a widely used technique, requires $N$ forward passes, typically 20 to 30, to approximate the underlying distribution. This leads to a computational cost roughly 30 times higher than our method, rendering it impractical for most systems.
>
> Our primary goal is to improve the control quality of diffusion policies **in real-world environments**, where maintaining a sufficient control rate, typically 30Hz as stated earlier, is critical. From this perspective, while probabilistic metrics may offer more precise evaluation, they are often infeasible in practice due to their heavy computational demands.
>
>
>
>
> ## Question 3
> > Does fixed-threshold chunking fail in edge cases (e.g., sudden disturbances)?
>
> Thank you for the insightful question. As shown in Figure 6, our method with a fixed threshold actually maintains strong and consistent performance across a wide range of tasks, highlighting its practical robustness to disturbance.
>
> However, we agree that a fixed threshold may not be theoretically optimal for all scenarios, especially where noise levels differ significantly between phases of a single rollout. Therefore, we believe that exploring a dynamic thresholding mechanism, which could adapt based on task context or uncertainty, is an interesting and valuable direction for future work.
>
> --- references ---
> [1] Liu, Yuejiang, et al. "Bidirectional decoding: Improving action chunking via closed-loop resampling.", ICLR 2025
> [2] Black, Kevin, et al. " $\pi_0$: A Vision-Language-Action Flow Model for General Robot Control."
> [3] Bjorck, Johan, et al. "Gr00t n1: An open foundation model for generalist humanoid robots." arXiv preprint arXiv:2503.14734 (2025).
> [4] Węglarczyk, Stanisław. "Kernel density estimation and its application." ITM web of conferences. Vol. 23. EDP Sciences, 2018.

---

> ### Comment · Area_Chair_D5ed · 2025-08-09
> **Comments by AC**
>
> Dear Reviewer,
>
> Thank you for your participation in the review process. If you haven't done these steps, please engage in the discussion phase by following these guidelines:
>
> - Read the author rebuttal;
> - Engage in discussions;
> - Fill out the "Final Justification" text box and update the "Rating" accordingly.
>
> Reviewers must participate in discussions with authors before submitting “Mandatory Acknowledgement”. The deadline is Aug 8, 11.59pm AoE.
>
> Thanks,
>
> AC

---

### Official Review · Reviewer_qBvg · 2025-07-04

**Clarity:** 4
**Significance:** 3
**Originality:** 3
**Rating:** 5
**Confidence:** 3

**Summary:**

The authors investigate means of improving generative behaviour cloning policies, particularly the diffusion policy approach. They propose that one issue with these models is that they do not function as “closed-loop” policies, in which decisions are made at each state to complete a task. To address this limitation, they propose a novel self-guidance technique using subsequent states to modify the noising process during action generation. They then propose an adaptive mechanism to decide when the current action sequence differs too much from the model's output and to either continue executing the current plan or use the new action sequence. Experimentally, their approach improves control performance over baseline methods that propose alternative means of adapting action sequence generations with diffusion policies, showing strong performance improvements.

**Questions:**

- Line 119: Why is “h = H / 2”?
- How fast do we need to be able to make a decision? Even in robotics, where sampling rates could make some difference
- What is “P” in Figure 3 precisely? It says it’s noise levels, but is that referring to the standard deviation of the noise process or something else?

**Ethical Concerns:**

["NO or VERY MINOR ethics concerns only"]

**Final Justification:**

I maintain my original score, which was to accept the paper with some revisions as we discussed during the rebuttal which the authors addressed.

**Limitations:**

We felt the author's work could benefit from a more extensive limitations discussion. Computation time for their approach seems a much more important issue than they allocate space for.

**Paper Formatting Concerns:**

None we saw

**Quality:**

3

**Strengths And Weaknesses:**

Strengths:

-  Claimed contributions are clearly stated in the introduction, making it easier to understand the objectives of the paper
- The proposed techniques are accessible and appear to require minimal changes to utilize Diffusion policies more effectively during deployment.
- Experiments demonstrate both real-world results and simulation outcomes. The authors conduct ablation experiments to put the mechanisms proposed in the paper into context and assess their effectiveness and utility. Inclusion of sensitivity results

Weaknesses:

In terms of the method techniques proposed and experiments conducted, the only two major concerns we had were about the following:
- No experiments were done to help motivate determine the theoretical conclusions shown in Equations 12 - 13. This intuition of “predicting into the future” seems testable where instead of using the subsequent steps, the authors could use varying distal states to calculate the $\epislon_{new}$, e.g. instead of $s_{t - \delta t}$ use something like $s_{t - k \delta t}$ where $k$ is varied over.
- Experiments considering the tradeoffs in computation by using a “closed-loop” approach. The authors briefly mention these limitations, noting that their approach may be slower if queried more frequently. If the authors performed experiments to demonstrate this tradeoff in computation time to previous approaches, it would put the consequences of their techniques' tradeoffs into perspective between performance and computation, particularly for real-world robotic use cases.

Other concerns we had which are more about how to position the work:
- We did not find the motivations for closed-loop vs. open-loop compelling justifications for the work. The horizons the author mentions (H = 16 steps) seem short, particularly if there’s a high sampling rate on a robot. If your system works at 1000 Hz, what is the difference in precision you’re losing at such a short window of prediction? As the authors point out in their limitations (line 334) regarding computation costs, the bigger issue, in our opinion, is not being able to query the model fast enough for real-time control in a closed loop system. To our understanding, diffusion is a fundamentally iterative process, so generating a single action chunk could take more time than possibly hope to query in between decisions.  We suggest that the authors focus their paper more on discussing the improvement of diffusion policy inference than on making claims about the benefits of being  a closed vs. open-loop control system, becuase we do not agree that their results support this argument well.

- The work should acknowledge fields addressing similar problems more beyond the diffusion policy research literature. In our opinion, the “adaptive action chunking” mechanism is similar in motivation to the stopping function defined in the options framework [1]. Likewise, fields like control theory have also studied the value of replanning action sequences despite being able to roll out multiple steps (e.g., model predictive control predicts multiple time-steps but typically only sends one command). It’s surprising, for instance, that lines 38-45 provide  no supporting research on the tradeoffs between “open loop” and “closed loop" approaches. We would appreciate the authors acknowledging that these motivations of their work are also present in other domains. In our opinion, these ideas (OL vs CL, stopping functions) are not novel by themselves, but instead addressing these issue specifically in the context of diffusion policies is the real focus.

[1] Sutton, Richard S., Doina Precup, and Satinder Singh. "Between MDPs and semi-MDPs: A framework for temporal abstraction in reinforcement learning." Artificial intelligence 112.1-2 (1999): 181-211.

---

> ### Author Rebuttal · Authors · 2025-07-30
>
> Thank you very much for your positive review and acknowledging the contributions of our work. We have carefully considered each of your points and present our detailed responses below.
>
> ## Weakness 1
> > No experiments were done to help motivate determine the theoretical conclusions shown in Equations 12 - 13 ... The authors could use varying distal states to calculate the $\epsilon_{new}$ , e.g. instead of $s_{t-\delta t}$ use something like $s_{t-k\cdot t}$ where $k$ is varied over.
>
> Thank you for your insightful and constructive feedback. In response to your suggestion, we revised our algorithm to incorporate varying internal states and evaluated its predictive quality by computing the mean $L_2$ distance between the future actions generated by SG and the ground truth in the Push-T environment. Specifically, we experimented with the formulation :
>
> $$
> \epsilon_{\text{SG}} \leftarrow (1 + w)\epsilon_{\theta}(x \mid s_t) - w\epsilon_{\theta}(x \mid s_{t - k})
> $$
>
> across different values of $k$, to assess how past states influence prediction accuracy.
>
> | | k=1 | k=2 | k=3 | k=4 | k=5 | baseline ($a_t - a_{t-1}$) |
> |---|---|---|---|---|---|---|
> | $L_2$ | **8.164** | 9.616 | 10.757 | 11.389 | 12.338 | 10.667 |
>
> We denote the baseline $\|a_t - a_{t-1}\|$ as the natural variance between consecutive ground-truth actions. As shown in the results, our SG method with $k = 1$ achieves the lowest $L_2$ error (8.164), outperforming the baseline (10.667) in predicting future actions. Moreover, the performance gap widens as the temporal offset $k$ increases, indicating that the accuracy of SG degrades more slowly over time compared to the baseline.
>
> This behavior suggests that SG effectively acts as a discretized approximation of continuous-time action trajectories. As the temporal gap decreases, the prediction becomes more accurate, highlighting the model's strength under locally linear dynamics. These results provide strong empirical evidence for the effectiveness of our extrapolation mechanism, and we will include this experiment in the final version of the paper.
>
>
> ## Weakness 2
> > Experiments considering the tradeoffs in computation by using a “closed-loop” approach.
> > If the authors performed experiments to demonstrate this tradeoff in computation time to previous approaches, it would put the consequences of their techniques' tradeoffs into perspective between performance and computation, particularly for real-world robotic use cases.
>
> Thank you for your valuable feedback. As you rightly noted, the trade-off between performance and computational cost is a critical concern in real-world robotics. In this regard, we would like to emphasize that **our proposed method offers a more favorable performance-computation trade-off than prior approaches \[1]**, as also demonstrated in our real-world experiments:
>
> * **Closed-Loop Latency:**
>   Similar to prior work \[1], our method assumes that modern GPUs can support closed-loop control without compromising standard visuomotor control frequencies (\~30Hz). In our real-world experiments using a 10-step diffusion policy on an A6000 GPU, the closed-loop controller comfortably maintains the \~30Hz requirement. We clarify that **the main contribution of our work lies in maximizing control accuracy under this assumption**, and we validate this through extensive empirical results.
>
> * **Performance-Computation Trade-off:**
>   Importantly, we highlight that **our method is significantly more efficient than prior methods such as AG \[6] and BID \[1]**. These approaches require multiple inferences to stabilize diffusion control, resulting in a high computational load that often **exceeds the practical limits of the assumed hardware**, leading to reduced control frequencies. As detailed in Appendix. I, our main baseline BID \[1] incurs \~16× higher computation, reducing control frequency to \~15Hz and causing unstable behavior and lower task success rates. In contrast, our method sustains stable 30Hz control without such issues.
>
> Your comment prompted us to reflect more deeply on how to better highlight one of the core contributions of our work. We sincerely appreciate the insightful suggestion and will incorporate this point into the final version of the paper.
>
> ## Weakness 3
> > We suggest that the authors focus their paper more on discussing the improvement of diffusion policy inference than on making claims about the benefits of being a closed vs. open-loop control system.
>
> Thank you for your insightful questions and valuable suggestions regarding the paper's position. As you pointed out, a more impactful contribution of our work is effectively improving diffusion policy inference, with our novel self guidance technique. We will be sure to emphasize this contribution more explicitly in the final version of the paper.
>
> > We did not find the motivations for closed-loop vs. open-loop compelling justifications for the work..
>
> However, we wish to reiterate that the contribution regarding the OL vs. CL framework is also critical contribution, especially towards improving a robot's real-world performance. Below are reasons and our responses to your specific questions.
>
> >If your system works at 1000 Hz, what is the difference in precision you’re losing at such a short window of prediction?
>
> Instead of low level control at ~1000Hz, our work mainly focuses on visuomotor policies,  which take entire camera images as input to decide on the next action (e.g., target end-effector coordinates). These policies conventionally operate at ~30Hz, which is a standard setting in this research domain. In this context, a prediction horizon of H=16 corresponds to approximately 0.5 seconds (16/30s) and is not a short-term window.
>
> In manipulation tasks, 0.5 seconds is a significant duration where environmental changes can cause task failure. While open-loop control is the de facto standards for modern generative model-based control, research into its risks has been limited explored [1].
>
> >...the bigger issue, in our opinion, is not being able to query the model fast enough for real-time control in a closed loop system.
>
> As you rightly pointed out, the inference speed of diffusion models is a major concern. However, as answered in Weakness 2, we operate on the assumption that modern systems are powerful enough to maintain a stable ~30Hz control loop. We believe this trend will continue and extend to edge computing, thanks to advancements in hardware and model optimization techniques.
>
> **Remark :** The challenge for closed-loop control we want to solve in our paper is no longer speed, but rather it's low performance despite their theoretical superiority by its lack of consistency and long-term planning abilities. Solving this problem is especially critical in real-world environments, where can be noisy or on low-cost robot systems. Our work directly tackles this issue to increase the performance (accuracy) of diffusion-based control systems.
>
> ## Weakness 4
> > The work should acknowledge fields addressing similar problems more beyond the diffusion policy research literature.
>
> > We would appreciate the authors acknowledging that these motivations of their work are also present in other domains.
>
> First, thank you for your valuable insights and feedback. We agree that the concepts of open-loop and closed-loop control are well-explored topics in fields such as Model Predictive Control,  Receding Horizon [2] and Options framework [3].
> Moreover, within the context of diffusion model-based control, prior studies have also investigated these concepts. For instance, BID [1] analyzes the performance trade-offs between open-loop (OL) and closed-loop (CL), while [4] provides a theoretical analysis of the effect of the horizon. We will incorporate these acknowledgments more clearly and clarify our main contributions.
>
> ## Questions
> >Line 119: Why is “h = H / 2”?
>
> - This follows standard of diffusion policy model [5] for fair comparison. We will add more description.
>
> > How fast do we need to be able to make a decision?
>
> - In the visuomotor policy tasks we focus on, the standard control frequency is typically around 30Hz. This corresponds to an action inference time of approximately 260ms in the open-loop setting (h=8) and 33ms in the closed-loop setting. As noted in the paper, such latency requirements are well within the capabilities of modern GPU-based systems.
>
> > What is “P” in Figure 3 precisely?
>
> - We denotes $P$ as a environmental noise, which is the speed of movement of T block and is the same noise setting in previous works, BID [1]. Please see Appendix.A also. We will clarify in final version.
>
>
> --- references ---
> [1] Liu, Yuejiang, et al. "Bidirectional decoding: Improving action chunking via closed-loop resampling.", ICLR 2025
> [2] Mattingley, Jacob, Yang Wang, and Stephen Boyd. "Receding horizon control." IEEE Control Systems Magazine 31.3 (2011): 52-65.
> [3] Option discovery using deep skill chaining. 2019
> [4] Foster, Dylan J., Adam Block, and Dipendra Misra. "Is behavior cloning all you need? understanding horizon in imitation learning." Advances in Neural Information Processing Systems 37 (2024): 120602-120666.
> [5] Chi, Cheng, et al. "Diffusion policy: Visuomotor policy learning via action diffusion." The International Journal of Robotics Research (2023): 02783649241273668.
> [6] Karras, Tero, et al. "Guiding a diffusion model with a bad version of itself." Advances in Neural Information Processing Systems 37 (2024): 52996-53021.

---

> > ### Comment · Reviewer_qBvg · 2025-08-04
> > **I still disagree on emphasizing closed loop vs open loop, but otherwise comments are addressed**
> >
> > At this point, I think it's just arguing semantics. I do not agree with the authors stances to focus on "closed loop vs open loop" but otherwise feel comments were adequately addressed. For my clarification, are the authors actually waiting for action chunks to finish (i.e. that they are waiting 0.5 seconds before sending another command) or blending actions at each time step? If it's the latter, then when exactly is the "open loop" control happening? If you're constantly adapting actions, this to me comes off as a closed loop system which is why I find this to be fundamentally weak motivation for the paper.
> >
> >  At a minimum, please clarify these details (the 30 Hz rate of control, what a horizon of 16 means in practice, etc.) in your paper as none of the arguments brought forth were obvious when reading the original draft.

---

> > > ### Author Response · Authors · 2025-08-05
> > > **Response to Reviewer qBvg**
> > >
> > > Thank you very much for engaging in the discussion and for your valuable feedback. We are pleased that we were able to address your concerns.
> > >
> > > Acknowledging your insightful suggestions, **we will surely incorporate the following clarifications into the final version of our paper**:
> > >
> > > -   Detailed description of the assumed control system (e.g., ~30Hz control, open-loop physical time, latency of Diffusion Policy (DP) on a modern GPU, etc.).
> > > -   Further emphasis on our contribution to improving DP, particularly through guidance.
> > > -   Numerical experiment on the future-prediction behavior of Self-Guidance.
> > > -   Expanded discussion and related work in action chunking beyond DP, such as the MCP, RHC, and Options frameworks.
> > >
> > > We also believe these will greatly aid in understanding the paper's motivation and contributions. We deeply thank you for your insightful suggestions.
> > >
> > > Furthermore, to clarify your question: it is the **former case**. In the open-loop control we used, actions within the $h$ horizon are executed without adjustment, and replanning(model inference) occurs only after the entire chunk has finished.

---

### Note · Authors · 2025-08-14

Dear AC and Reviewers,

We sincerely thank you for your time and constructive evaluation. We are encouraged that you found our method to be **innovative and simple** (eCfX, fbxt, qBvg, tuvk), noted its **significant performance improvement** (eCfX, qBvg, tuvk), and appreciated the **extensive experiments conducted in both simulation and real-world settings** (eCfX, fbxt, qBvg).

Below, we summarize the key points clarified during the rebuttal:

***

1.  **Broad Applicability to recent VLA Methods:**
    We provided a richer evaluation by applying our method to recent SOTA Vision-Language-Action (VLA) models, $\pi_0$ and GR00T-N1, in both simulation (LIBERO) and real-world experiments, to address questions noted by Reviewers (tuvk, fbxt, eCfX). The results demonstrated **significant performance gains, confirming the broad applicability of our method**.

2.  **Empirical Validation of Self-Guidance's Future Prediction:**
    We conducted a new quantitative experiment to explicitly measure the alignment between Self-Guidance (SG) predictions and actual future actions, addressing questions from Reviewers (qBvg, fbxt). The results show that SG-predicted actions have a substantially lower error, validating that **SG effectively guides the sampling process with approximated future information** by leveraging a linear approximation of past states.

3.  **Novelty Against Autoguidance (AG) and BID:**
    We presented a detailed ablation study directly comparing individual components of our method with AG/BID to clarify its novelty and advantages, addressing a question from Reviewer (tuvk). The results demonstrate that **our method achieves the highest performance** across all settings. We also clarified a key difference: our method has a **significantly lower computational cost** than these methods.

4.  **Robustness of the Method:**
    We provided further analysis of our method's performance in challenging scenarios, including "slowing down" and "mode-toggling," to address potential failure cases raised by Reviewer (eCfX). We demonstrated that our method **does not falter in these scenarios** and showed that its components **work synergistically to ensure robust performance**.

5.  **Other Clarifications:**
    We will also address all suggestions raised during the discussion—such as adding more references, providing a detailed description of the control system, and clarifying the definition of $P$—in the revised manuscript to enhance its overall quality.

---

### Decision · Program_Chairs · 2025-09-17

**Decision:**

Accept (poster)

**Comment:**

The paper received ratings of 4, 5, 5, 5. After rebuttal, reviewers acknowledged that the authors had provided extensive new experiments—covering both simulation (LIBERO) and real-world robotics, as well as additional VLA models (e.g., GR00T-N1)—and clarified theoretical and computational tradeoffs. Reviewer eCfX raised the rating to Accept (5). Reviewer tuvk raised the rating to Borderline Accept (4) after recognizing the broad applicability and robust empirical validation of the proposed methods. Reviewer qBvg maintained an Accept (5), noting the paper’s contributions to diffusion policy inference.

Overall, the proposed Self-Guidance and Adaptive Chunking modules are simple, plug-and-play, and training-free techniques that yield consistent improvements across a wide range of robotic manipulation tasks. While some questions about conceptual novelty remain, the rebuttal and follow-up results sufficiently addressed the major technical concerns. I recommend Accept.